# Envision the Future in Open-World Dynamic Tasks by a Hierarchical World Model with Residual Enhanced Foresight

## Abstract

Interacting with proactive agents in open-world environments remains a core challenge for reinforcement learning, as other participants exhibit reciprocal and even adversarial behavior. Effective reasoning representations are crucial in such settings. Although language- or vision-grounded approaches have shown promise, most depend on large-scale pre-training and extensive domain-specific fine-tuning. Drawing inspiration from the neuroscience phenomenon of active gaze control—where humans proactively direct gaze toward the predicted future location of a dynamic object (e.g., a cricket ball, blade, or oncoming vehicle) well before distinguishing visual features appear—we propose ResDreamer, a brain-inspired hierarchical world model that employs residually connected visual planning representations. In ResDreamer, each higher-level layer learns on the reconstruction residuals of the layer below, enabling progressive capture of increasingly advanced world dynamics and the construction of a richer internal representation. Every layer incorporates augmented observations that include foresight images that are further modulated by top-down residual prediction signals. This mechanism yields highly informative, predictive, and knowledge-driven visual reasoning representations without external supervision. Empirical results demonstrate that ResDreamer achieves higher sample efficiency, parameter efficiency, and scalability compared to state-of-the-art baselines, paving the way for more adaptive agents in open-ended, dynamic, and interactive environments.

## 1 Introduction

In interaction or combat scenarios, task objectives are relevant with dynamic elements or even another proactive agent. This introduces a highly dynamic and uncertain environment evolution. The vast state space of an open-ended environment exacerbates this challenge. The agent must construct an internal world representation based on partial information and make decisions accordingly.

Task planning representation is critical for achieving human-level intelligence in open-ended dynamic environments. World model has pushed the boundaries of reinforcement learning (RL) (Schrittwieser et al., 2020; Robine et al., 2023; Zhang et al., 2023; Alonso et al., 2024). DreamerV3 (Hafner et al., 2025) achieved high performance generalization across 150 diverse tasks with a unified hyperparameters. However, most existing model based RL (MBRL) methods only consider pixel reconstruction as one of the gradient signals for representation learning during training. The decision making process did not directly benefit from the world model's ability to predict future sensory signal.

Hierarchical methods can naturally support task planning by transmitting plan representations between layers. Strategies at different levels allow for independent optimization on different timescales (Lee et al., 2022; Gumbsch et al., 2024). The latent space targets can provide guidance for the lower-level worker (Hafner et al., 2022; Vezhnevets et al., 2017). JARVIS-1 (Wang et al., 2023b), MC-Planner (Wang et al., 2023c), and RL-GPT (Liu et al., 2024) are recent open-ended embodied agents that integrate RL with Large Language Models (LLMs). Leveraging their generalized world knowledge, LLMs can provide advanced, interpretable language-based task plans through approaches such

as task decomposition and policy-as-code. However, low-frequency target may become infeasible when the task relevant objects are in dynamic moving and active states.

We hold the view that multi-level reasoning is crucial for open-world general intelligence. Therefore, an ideal planning representation should hierarchically capture world dynamics at different levels of abstraction. Neuroscience evidence suggests that the biological neural signals encode prediction error rather than raw sensory input (Rao & Ballard, 1999; Hosoya et al., 2005). Visual neurons employ a dynamic predictive coding strategy to filter out predictable components from the visual stream, transmitting only unexpected surprise or "report valuable" stimuli (Kok & de Lange, 2015).

Based on the above insights, we present ResDreamer, a hierarchical world model with residually connected visual planning representations. The residual of visual reconstruction serves as a signal for interlayer interaction in hierarchical world models. Information about prediction errors and feedback is transmitted between layers without the propagating gradients. The higher-level world model, by modeling visual residuals, not only constructs a comprehensive internal representation of the world but also refines the lower level's predictions through residual reasoning, thereby providing more accurate foresight.

In summary, the major contributions of this work are:

- We propose a general hierarchical architecture for world models, in which enhanced visual observations enable brain-inspired bidirectional transmission of foresight predictions and sensory surprises between adjacent layers, paving the way for scalable world models to enter the "ResNet era" of RL.

- Experimental results validate the sample efficiency, parameter efficiency and scalability of our approach in online RL context. Through ablation studies, we demonstrate that both foresight rollout and residual modulation contribute to the performance gains.

## 2 RELATED WORK

**MBRL**. Recurrent world dynamic models facilitate representation, simulation and policy improvement in MBRL (Ha & Schmidhuber, 2018). MuZero (Schrittwieser et al., 2020) conducts Monte Carlo tree search in the latent space by the learned state space model. DreamerV3 (Hafner et al., 2025) outperformed expert models tuned for specific domains and, for the first time, successfully collected diamonds from scratch in Minecraft. LS-Imagine (Li et al., 2024) breaks the limitations of single-step reasoning and uses the affordance map to trigger the cross-step jump prediction. It simulates jumping to the vicinity of high return targets in the future by magnifying specific areas in the observed image. In visual MBRL, transformer-based architectures (Micheli et al., 2022; Robine et al., 2023; Zhang et al., 2023) and diffusion models (Alonso et al., 2024) have emerged as particularly effective paradigms for world modeling, offering enhanced expressivity and sample efficiency. However, as far as we know, there is no MBRL method that naturally builds a hierarchical representation learning architecture based on the reconstruction residuals of sensory signals.

**Hierarchical RL**. Hierarchical RL is considered promising in alleviating the exploration stagnation caused by sparse rewards in complex and long-term tasks. Capturing task-relevant details across varying temporal scales is a key focus of current research efforts (McInroe et al., 2022; Rao et al., 2023; Schiewer et al., 2024; Lin et al., 2024; Li et al., 2024). Beyond static temporal scales, THICK (Gumbsch et al., 2024) adaptively discovers larger temporal scales by guiding lower-level world models to sparsely update their partial latent states. Automatic goal discovery is another critical aspect, enabling agents to autonomously identify and pursue meaningful objectives (Hafner et al., 2022; Hamed et al., 2024; Nicklas Hansen, 2025). Puppeteer Hansen et al. (2024) directly uses expert trajectories from human MoCap data to train high-level goals for lower-level whole-body humanoid controller. Hierarchical methods can benifit from internet-scale datasets to provide generalized prior knowledge for the lower-level policy (Baker et al., 2022; Yuan et al., 2024; 2023). However, none of the existing method exchange visual residual signal between layers. In our approach, higher-level models can be scalably stacked to achieve increasingly comprehensive representation learning.

**World model**. Recently, Vision-Language Model (VLM) (Cen et al., 2025) and Joint Embedding Predictive Architecture (JEPA) (LeCun, 2022) have emerged as competitive world model architec-

tures. Web-scale pre-training and relatively sufficient expert demonstration are often prerequisite of VLA driven Minecraft agents (Wang et al., 2023a;c; Li et al., 2025). JEPA is a self-supervised representation learning framework. As an autoregressive generative architecture, it is pretrained on Internet scale multimedia data in the absence of pixel-level reconstruction. Various instances of JEPA have demonstrated its potential across a wide range of domains, including images (Assran et al., 2023), videos (Bardes et al., 2024; Assran et al., 2025), optical flow (Bardes et al., 2023), point clouds (Saito et al., 2025), and graph data (Skenderi et al., 2023). Recent works have shown that hierarchical world models representation learning is more beneficial for downstream tasks Mounir et al. (2023); Rao et al. (2023). However, the hierarchical representation architecture still requires further exploration on a wider range of tasks. We propose a hierarchical world model that requires no domain-specific knowledge. It naturally supports the incorporation of internet-scale knowledge, such as the temporal reward signals provided by MineClip (Fan et al., 2022).

# 3 METHOD

In this section, we present the details of ResDreamer. We introduce ResDreamer from the perspectives of representation learning and behavior learning. First, we describe the basic module of each layer in our Hierarchical Recurrent State-Space Model (HRSSM). Next, we present our primary innovation in representation learning architecture, namely the enhanced observation through residual connection. Finally, we formalize the loss functions and the overall training algorithm.

## 3.1 HIERARCHICAL WORLD MODEL

We implement the HRSSM based on Predictive Processing Blocks (PPBs). Predictive Processing or Predictive coding is a paradigm to explain hierarchical reciprocally connected organization of the cortex (Huang & Rao, 2011).

In the k-th layer block $\text{PPB}^k$, recurrent state contains the deterministic state $h_t^k$ and the stochastic state $z_t^k$. The sequence model is used to represent the state transitions conditioned by action taken. The Encoder extracts useful information from the new input observations to guide the recurrent state update, while the Predictor attempts to predict the stochastic state without accessing the observations.

$$\text{PPB}^k \begin{cases} \text{Sequence model:} & h_t^k = S_\phi\left(z_{t-1}^k, h_{t-1}^k, a_{t-1}\right) \\ \text{Encoder:} & z_t^k \sim q_\phi\left(z_t^k \mid h_t^k, o_t^k\right) \\ \text{Predictor:} & \hat{z}_t^k \sim p_\phi\left(\hat{z}_t^k \mid h_t^k\right) \\ \text{Decoder:} & \hat{o}_t^k \sim D_\phi\left(\hat{o}_t^k \mid h_t^k, z_t^k\right). \end{cases} \quad (1)$$

where $\hat{z}_t^k$ is the predicted stochastic state, $o_t^k$ and $\hat{o}_t^k$ are true and reconstructed observations. Layer index $k = 0, 1, \cdots L - 1$ and $L$ is the number of HRSSM layers. Each layer's PPB module contains all the components of the dreamerV3 (Hafner et al., 2025).

## 3.2 VISUAL HINT STRUCTURE AND RESIDUAL MODELING

Figure 1 gives an overview of ResDreamer, a hierarchical world model in which layers communicate through error feedback and predictive visual hints. This section elaborates on the forms of information exchange between the world model layers. ResDreamer is characterized by progressive residual learning of sensory signals and image foresight corrected by residual prediction.

Sensory reconstruction error is fed into the higher-level world model for residual learning. The **lower residual observation** is given by

$$o_{\text{res}}^k = \begin{cases} \text{empty set,} & k = 0, \\ \text{Norm}^k\left(o_{\text{raw}} - \hat{o}_{\text{raw}}\right), & k = 1, \\ \text{Norm}^k\left(o_{\text{res}}^{k-1} - \hat{o}_{\text{res}}^{k-1}\right), & k = 2, 3, \cdots, L - 1. \end{cases} \quad (2)$$

where $o_{\text{raw}}$ and $\hat{o}_{\text{raw}}$ are original and reconstructed environmental input. The omitted time indices are all $t$, and the same applies hereafter. $\text{Norm}^k(\cdot)$ computes the mean and variance across the pixel

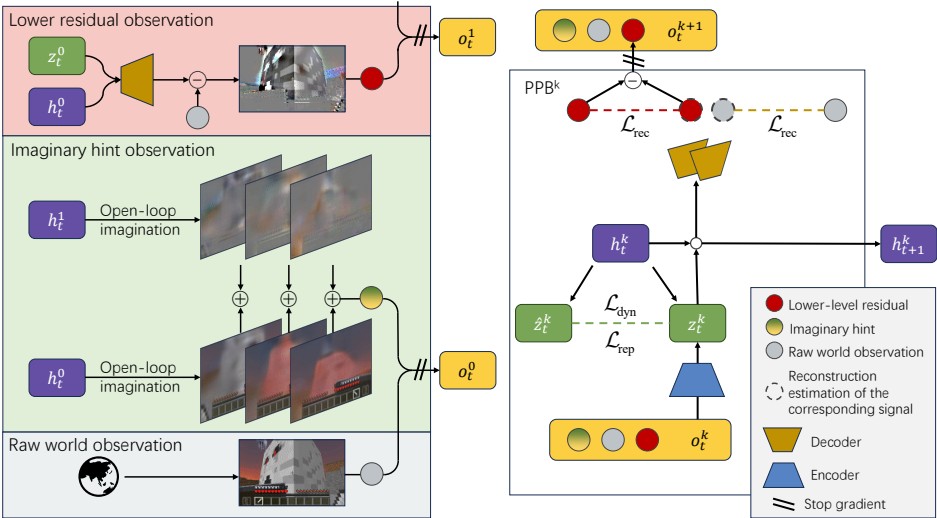

Figure 1: Overview of ResDreamer a model base RL algorithm based on hierarchical world model. The left side shows the structure of enhanced visual observations. Adjacent world model layers communicate by residual and predictive signal within the enhanced observation. The right side shows the modules and training process of the k-th layer world model. The Encoder reads enhanced visual observations and gives the posterior $z_t^k$. The dynamic predictor learns to estimate $z_t^k$ with $\hat{z}_t^k$ without accessing the observation. The sequence model updates internal state $h_t^k$ by $z_t^k$. The Decoder reconstructs the observation signal which generates reconstruction loss and residual visual signal for upper layer.

dimension and updates them using an exponential moving average. The **lower residual observation** only needs to call the decoder during the observation preprocessing stage, without the need to introduce any additional modules.

It is worth noting that any layer of the well-trained PPB can rollout latent trajectories by replacing the posterior with the prior. This means that as long as PPB is trained to model the lower residual, it can perform open-loop reasoning and correct the visual reasoning representations at the lower level. The **imaginary hint observation** is given by

$$
o_{\text{imag}}^k = \begin{cases} \left\{ \hat{o}_{\text{raw}} + \hat{o}_{\text{res}}^1 \right\}_{t:t+H}, & k = 0, \\ \left\{ \hat{o}_{\text{res}}^k + \hat{o}_{\text{res}}^{k+1} \right\}_{t:t+H}, & k = 1, 2, \cdots, L-2, \\ \left\{ \hat{o}_{\text{res}}^k \right\}_{t:t+H}, & k = L-1, . \end{cases} \tag{3}
$$

where the subscript $\{\cdot\}_{t:t+H}$ stands for index $t, t+1, \cdots, t+H-1$. Specifically, if the raw image shape is $(h, w, 3)$, then $\{\hat{o}_{\text{res}}^k\}_{t:t+H}$ and $\{\hat{o}_{\text{imag}}^k\}_{t:t+H}$ decoded from a $H$ steps latent trajectory both have the shape of $(H, h, w, 3)$. They are added by residual connections and concatenated along the channel axis into shape $(h, w, 3H)$.

The **imaginary hint observation** utilities the HRSSM's reasoning capabilities to directly envision the future. From the perspective of convolutional neural networks (CNN), the imaginary hint effectively generates dynamic CNN kernels based on predictive visual foresight. This process is similar to "gaze control" in neurosciences, which refers to the fact that attention is determined by knowledge-driven prediction (Jovancevic-Misic & Hayhoe, 2009; Henderson, 2017).

The observation expanded by lower-level residuals and upper-level predictions is the key to our hierarchical world model:

$$
o_t^k = \text{sg}\left( \left\{ o_{\text{imag}}^k, o_{\text{raw}}, o_{\text{res}}^k \right\}_t \right). \tag{4}
$$

where $\text{sg}(\cdot)$ is stop gradient operation. In our implementation, all the observations are concatenated along the channel axis. Eventually, a complete enhanced observation tensor is formed with the shape of $(h, w, 3H + 6)$.

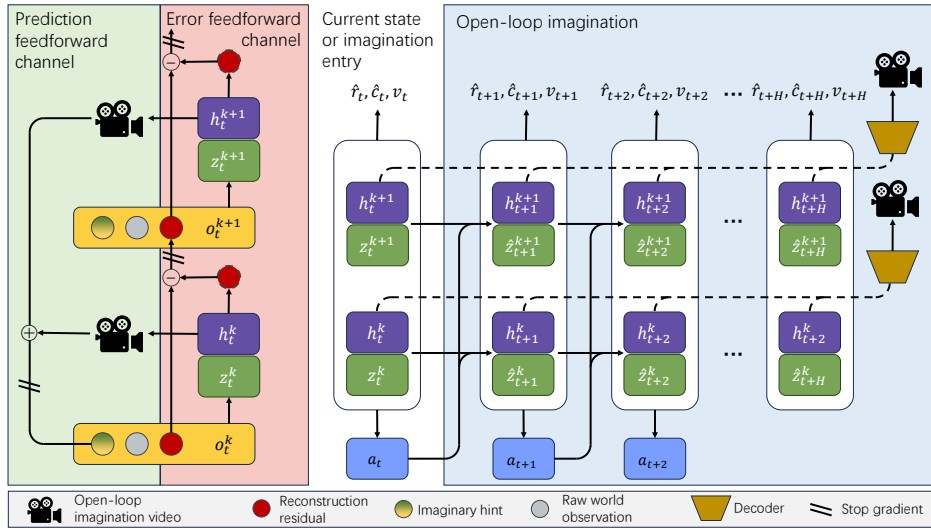

Figure 2: The information channel between world model layers is bidirectional. Only reconstruction error and modulated foresight images are transmitted between layers, with no gradients being passed. On one hand, each layer of the PPB generates predictions about the external world and transmits visual planning representations to lower layers. On the other hand, the PPB treats low-level residuals as self-supervised learning signals to obtain a more complete inner representation.

The **imaginary hint observation** is merely additional guidance or enhancement for the encoder. Only the **lower residual observation** and **environmental input** are reconstructed by the decoder and provide reconstruction loss during the training $\hat{o}_t^k = \text{sg}\left(\left\{\hat{o}_{\text{raw}}, \hat{o}_{\text{res}}^k\right\}_t\right)$.

Figure 3 shows the raw observation and imaginary hint observation on world model bottom layer while the agent is combating a ghast. The complete process of constructing enhanced observations from the bottom layer to the top layer and updating the recursive state in sequence is shown in Algorithm 1.

At this point, we have established the feedforward and feedback information channels of the hierarchical world model based on the enhanced observation (see Figure 2). This architecture combines the bandwidth advantage of inter-layer communication and the computational efficiency advantage within layers.

The visual hint does incur necessary computational cost, but from the perspective of parameter scale, the above architecture introduces almost no overhead. Within each layer, although the number of image channels has significantly increased due to the addition of video hint, the distribution of the visual hint is highly matched with the original image distribution benefits from the residual modeling, thus allowing for the sharing of major convolutional features. Therefore, in practice, we have not expanded the depth of the encoder and dimensions of stochastic state compared to dreamerV3 (Hafner et al., 2025).

## 3.3 WORLD MODEL AND BEHAVIOR LEARNING

In the context of RL, the final goal is to improve the policy. The actor-critic method is employed for policy optimization and state value learning.

$$\begin{aligned} \text{Actor:} \quad & a_t \sim \pi_\theta\left(a_t \mid s_t\right) \\ \text{Critic:} \quad & v_t \sim v_\psi\left(v_t \mid s_t\right) \end{aligned} \tag{5}$$

where $s_t = \left\{s_t^0, s_t^1, \cdots, s_t^{k-1}\right\}$ or its subset. Additional heads perform reward modeling $\hat{r}_t \sim p_\phi\left(\hat{r}_t \mid s_t\right)$ and episode-continuation flag prediction $\hat{c}_t \sim p_\phi\left(\hat{c}_t^k \mid s_t\right)$, which aid representation learning.

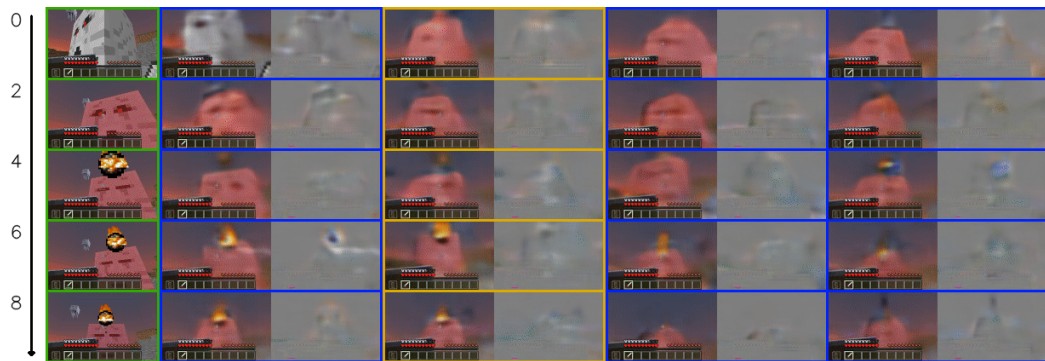

Figure 3: Visualization of Residual Enhanced Visual Observation on layer 0. It can be seen that at timestep 0, the agent had inferred in imagination that the opponent would turn red and enter the attack state at timestep 2, and drop the bomb at timestep 4. **Green**: raw observation. **Blue**: imaginary hint observation. **Yellow**: imaginary at time-step of next row. Each row shows the complete observation of the input encoder, with an interval of 2 timesteps. The video segment shows a ghast faces the agent and shoots a fireball. The agent reasons in imagination and makes a retreating evasive move.

The world environment generously provides continuous stream of sensory signals. Reconstructing sensory inputs serves as a critical training signal for world models. This drives the model to encode as much environmental information as possible in deterministic state.

$$\mathcal{L}_{\text{rec}}^k (\phi) = - \ln p_\phi \left( o_{\text{raw}} \mid s_t^k \right) - \ln p_\phi \left( o_{\text{res}}^k \mid s_t^k \right). \tag{6}$$

The stochastic state serves as the information channel through which new observation guide model updates. For instance, in a typical configuration of 32 categorical variable with 32 classes, the encoder extracts only 256 bits of information from observations at each step. Therefore, the encoder has to retain only the most critical information for updating the internal model. The enforced sparsity makes the stochastic state more feasible to predict, while the representation loss ensures that it tends to converge to a more predictable representation.

$$\mathcal{L}_{\text{dyn}}^k(\phi) = \max \left( 1, \text{KL} \left[ \text{sg} \left( q_\phi \left( z_t^k \mid h_t^k, o_t^k \right) \right) \parallel \quad p_\phi \left( z_t^k \mid h_t^k, \right) \right] \right)$$
$$\mathcal{L}_{\text{rep}}^k(\phi) = \max \left( 1, \text{KL} \left[ \quad q_\phi \left( z_t^k \mid h_t^k, o_t^k \right) \parallel \text{sg} \left( p_\phi \left( z_t^k \mid h_t^k \right) \right) \right] \right) \tag{7}$$

The prediction heads are similarly trained in a self-supervised manner, with the only difference being that they are conditioned on a stack of recurrent states from all layers.

$$\mathcal{L}_{\text{heads}}(\phi) = - \ln p_\phi \left( r_t \mid s_t \right) - \ln p_\phi \left( c_t \mid s_t \right) \tag{8}$$

Assuming that the world dynamic and the task-related experiences can be stably represented by the world model, the actor-critic can learn from the imaginary state trajectories, thereby significantly improving the sample efficiency.

The value distribution may span multiple orders of magnitude. Therefore, we parameterize the critic as a categorical distribution with exponentially spaced bins. We compute the bootstrapped $\lambda$-return $R_t^\lambda$ to train the critic. $R_t^\lambda$ accounts for $r_t$ within the trajectory horizon $T$ and incorporates the critic's expected value for returns beyond the horizon. The reward signal $r_t$ may originate from the environment or be estimated by the reward prediction head from imagined trajectories.

$$\mathcal{L}(\psi) = - \sum_{t=1}^{T} \ln p_\psi \left( R_t^\lambda \mid s_t \right)$$
$$R_t^\lambda = \begin{cases} r_t + \gamma c_t \left( (1 - \lambda) v_t + \lambda R_{t+1}^\lambda \right), & t < T \\ \mathbb{E} \left[ v_\psi \left( \cdot \mid s_t \right) \right], & t = T \end{cases} \tag{9}$$

The actor learns to maximize returns with entropy regularizer. To remain robust to outliers, we track the range between the 5th and 95th percentiles of returns using an exponential moving average. For further details on the loss function.

$$\mathcal{L}(\theta) = -\sum_{t=1}^{T} \frac{R_t^\lambda - \mathrm{sg}\left(v_\psi\left(s_t\right)\right)}{\max(1, S)} \log \pi_\theta\left(a_t \mid s_t\right) + \eta \mathrm{H}\left[\pi_\theta\left(a_t \mid s_t\right)\right]$$

$$S = \mathrm{EMA}\left(\mathrm{Per}\left(R_t^\lambda, 95\right) - \mathrm{Per}\left(R_t^\lambda, 5\right)\right)$$

(10)

We follow the official Dreamer V3 Hafner et al. (2025) hyperparameters and implementation details. Benefits from DreamerV3's robust generalization capabilities, ResDreamer can be trained across diverse tasks without hyperparameter tuning.

## 4 EXPERIMENTS

Engaging in combat within open-ended worlds presents significant challenges including terrain comprehension, the utilization of weapons and defensive tools, and dynamic anticipation of enemy movements. We evaluate ResDreamer on 5 combat tasks in MineDojo (Fan et al., 2022) as is introduced in Table 2.

In MineDojo tasks, the agent is equipped with iron armors and iron sword shield at initialization. We adopt sparse reward from MineDojo at episode termination and dense reward from MineCLIP (Fan et al., 2022). Each MineCLIP reward is computed of video segment of 16 time-steps, with calculations taking place every 8 frames. In addition, the agent is rewarded at any valid attack and punished for losing health points. The agent is trained for $1 \times 10^6$ environment steps. The image input of the MineCLIP model is $160 \times 256$ pixels, while ResDreamer observers 2x down-sampled images. All experiments can be reproduced with VRAM less than 29 GB.

The only non-trivial hyperparameter introduced by ResDreamer is the rollout horizon of image foresight. We provide sensitivity analysis of foresight horizon $H$ and time stride $D$ on DMC Vision Ortiz et al. (2024) tasks. DMC Vision is a visual control suite with continuous action space. We select the 5 most difficult tasks that Dreamer V3 did not converge to full score. We conducted all the "Dreamer" experiments under a unified set of hyperparameters, which verifies the cross-task generalization ability of ResDreamer.

### 4.1 MAIN COMPARISON

We measure the performance for all the methods with success rates during training. Our implementation is based on DreamerV3 (Hafner et al., 2025) and provides a brain-inspired hierarchical scaling method for it. To make a fair comparison with it in terms of parameter efficiency, ResDreamer is tested with two parameter configurations. Other baselines use the default configuration of the official implementation. Further details are provided in Appendix A.

The training curves in Figure 4 and comparisons in Figure 7 suggest that ResDreamer is an sample efficient hierarchical RL method. ResDreamer (100Mx2) adopts a 2-layer ResDreamer model with 100M parameters each layer. It demonstrates the best sample efficiency and convergence speed across all tasks. Despite using sparse hierarchical connections and only 84% of the parameter size, ResDreamer (50Mx2) has still surpassed the average performance of the DreamerV3. The detailed parameter size, hyper parameter and compute budget is shown in Table 1.

ResDreamer (100Mx2) is the only method that solves combating a shulker in $1 \times 10^6$ environment steps. The Shulker launches a guided projectile that causes prolonged levitation and causes fall damage afterwards. This poses complex and challenging dynamic interaction mechanisms for the agent. We found that the residual-modulated visual reasoning representation provided significant assistance in understanding this dynamic interaction mechanism. On the one hand, the feedforward channel provides modulated foresight image which makes the observation source move informative. On the other hand, the feedback channel allows error propagation upward, resulting in a more comprehensive inner world representation.

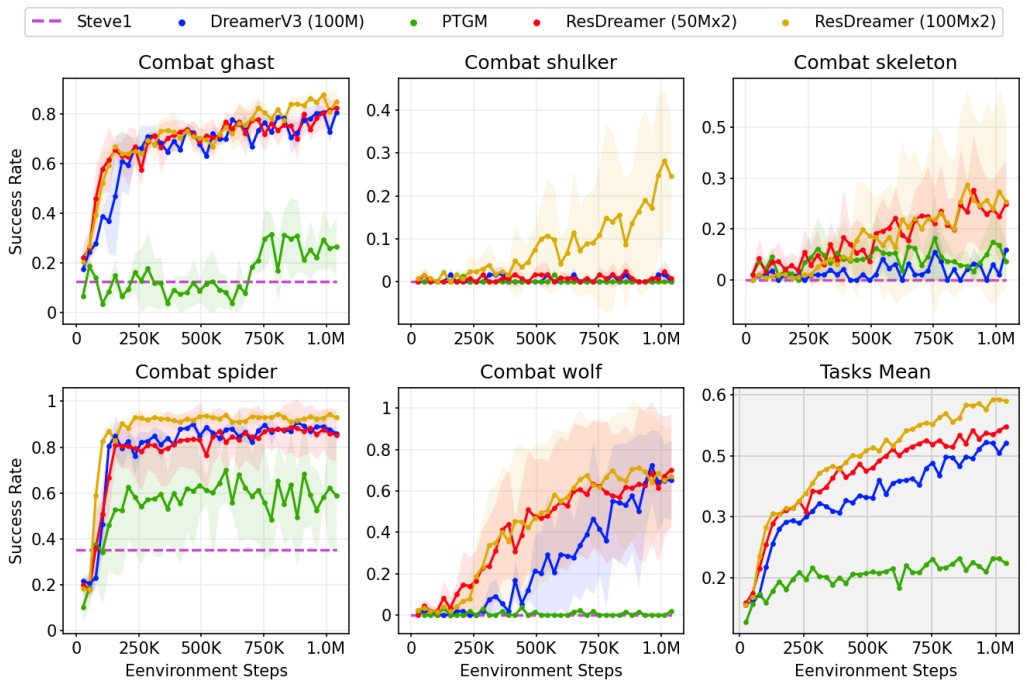

Figure 4: Comparison of ResDreamer against Steve-1 (Lifshitz et al., 2023), DreamerV3 (Hafner et al., 2025), PTGM (Yuan et al., 2024). We introduce the compared models in Appendix D.

## 4.2 MODEL ANALYSIS

The foresight rollout and residual connection mechanism are the key components of Res-Dreamer. We present ablation study to isolate the contribution from each design.

### 4.2.1 ABLATION STUDY

The enhanced observation through residual connection is a key feature of ResDreamer, enabling the flow of predictive and error information across the layers of the world model. Figure 5 show the results of the following alternative setups.

**ResDreamer (50Mx3)**: the ResDreamer model from the main comparison is extended to three layers. ResDreamer aims to learn more comprehensive world representations in a scalable manner. The mean task success rate of the three-layer ResDreamer surpasses that of the two-layer version. This indicates that ResDreamer provides an effective method for scaling up world models.

**ResDreamer (Heads conditioned on all)**: the actor, critic, and prediction heads in Res-Dreamer are conditioned on the recursive states of all layers. Experimental results show a performance decline under equivalent environ-

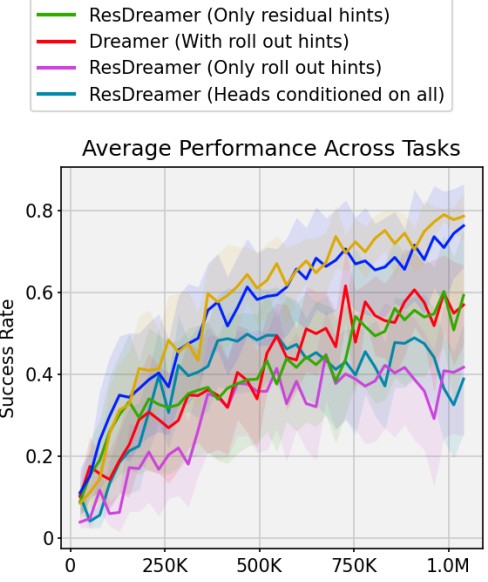

Figure 5: ResDreamer ablation study results.

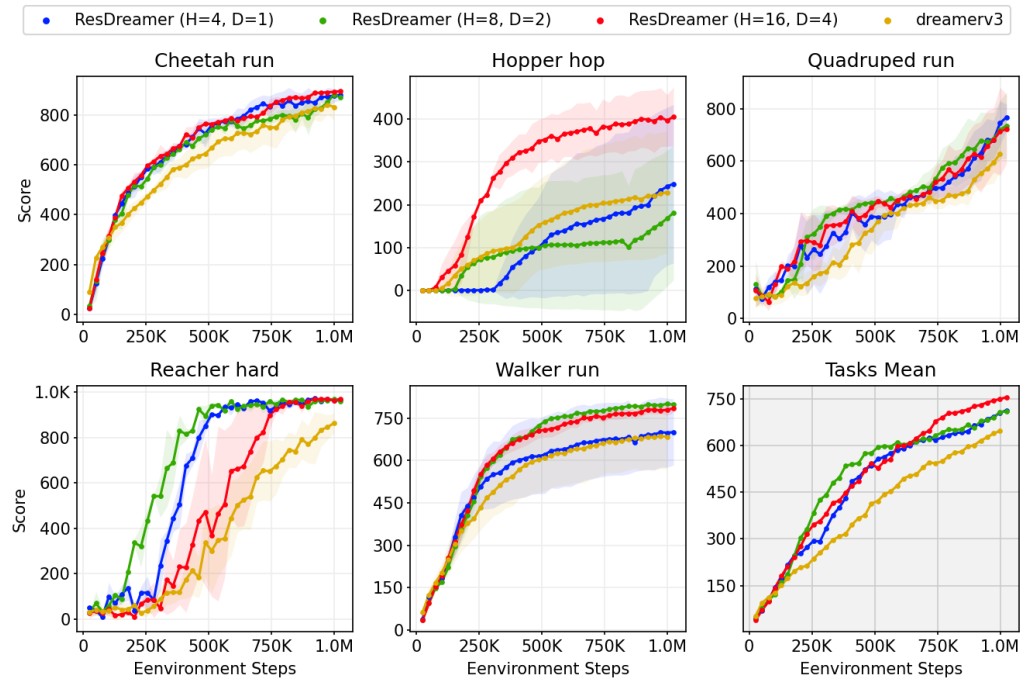

Figure 6: Comparison of ResDreamer with different foresight horizon on DeepMind Control Vision Ortiz et al. (2024) continuous control suite. The total number of frames for visual foresight has been aligned to 4 frames. Therefore, the horizon of $H = 8$ has stride $D = 2$, and $H = 16$ has stride $D = 4$.

mental interaction steps. Theoretically, complete recursive states contain more information, but the distribution of lower residual observations shifts during training process of lower-layer models, leading to relatively unstable representations in the upper-layer world model before convergence. Further studies could compare performance across additional environmental interaction step settings.

**ResDreamer (Only residual hints)**: the imaginary hint observations in ResDreamer consist solely of residual signals from the upper layer, without incorporating the current layer's predictive reconstruction. Although the current layer's recursive state already encompasses complete information from open-loop predictions, we find that imaginary hint observations with residual connections yield superior performance.

**ResDreamer (Only rollout hints)**: residual connections are removed compared with standard ResDreamer. Actor-critical and other prediction heads use the learned representation simply by accessing the latent vector class of all layers. The tasks performance drops which suggests that visual foresight modulated by residual rollout is more informative.

**Dreamer (With image foresight)**: a enhanced version of single-layer Dreamer V3, allowing it to obtain online computed visual grounded reasoning observation by foresight rollout and reconstruction. The performance drops which further verifies the contribution from hierarchical architecture and residual connection mechanism.

### 4.2.2 FORESIGHT HORIZON SENSITIVITY ANALYSIS

**ResDreamer (Rollout with horizon H=4, 8, 16 and stride D=1, 2, 4)**: different rollout horizon and time stride is tested on DMC Vision tasks Ortiz et al. (2024). The result in Figure 8 shows that longer horizon and larger stride eventually converge to higher average performance. However, on some tasks, a long horizon in the early stages of training can lead to slower convergence.

ResDreamer outperforms Dreamer V3 with aligned experiment setup. Generally, after sufficient training, foresight with longer horizon and larger stride is more informative, but the convergence speed may be slower in some cases.

In "DMC Reacher Hard" which requests precise control of a two-link robotic arm, all ResDreamer configurations converge to the full score within 1M steps. The reasoning table with a shorter time window converges the fastest.

The "DMC Hopper Hop" requires a 4-joint legged robot to perform a high-speed forward leap while maintaining balance. The agent with rollout horizon $H = 16$ significantly outperforms other setup for 16 steps roughly covers the entire cycle of a single jump.

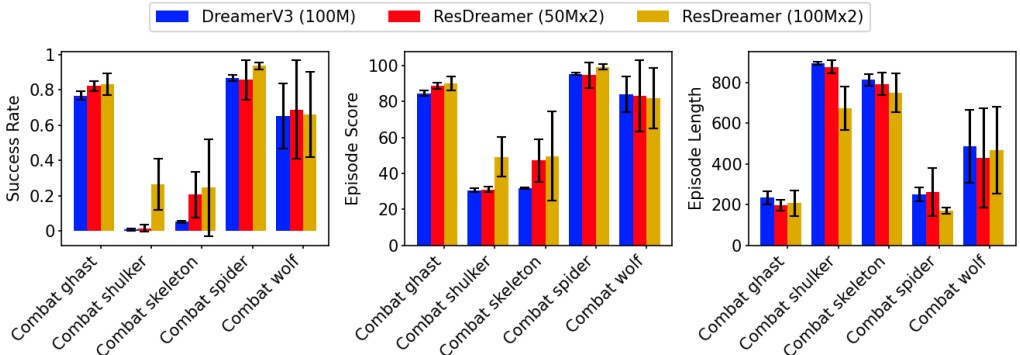

Figure 7: Comparisons of success rate (↑), episode score (↑) and episode length (↓) across tasks. It can be seen that ResDreamer achieves higher scores and success rates with fewer steps. Although the ResDreamer (50Mx2) has slightly fewer total parameters than DreamerV3 (100M), it performs better in almost all tasks.

## 5 CONCLUSION

In this paper, we present ResDreamer, a hierarchical world model featuring residual-connected visual planning representations. Residual enhanced observations establish an information channel between layers. Those explainable sensory signals are stripped, and the remaining novel stimuli are passed on to higher-level models for learning. The residual rollout from high-level modulates the visual grounded planning representation, helping the encoder perform gaze control based on more informative visual information source.

Through comparisons with baselines and model analysis, we demonstrate that ResDreamer achieves superior sample efficiency with fewer parameters compared to baselines. The combination of hierarchical structure, hint rollout, and residual modulation is significantly more effective than any subset of these components. In other words, residual-modulated image foresight is significantly more informative than any subset of it.

Central to the ResDreamer design is the enhanced observations that integrate lower residuals with higher level predictive foresight. This simple yet powerful principle underlies the model's ability to scale efficiently, supporting deeper hierarchies at the cost of only linear increases in communication bandwidth. ResDreamer facilitate excellent world model scalability. Data exchange occurs only between adjacent layers, and the communication bandwidth consumption increases linearly with the number of parameters.

The primary limitation of ResDreamer lies in its static image foresight horizon length. Long-horizon goal image increases computational cost, whereas overly short visual hints may fail to provide sufficient environmental dynamics information. We leave the development of adaptive-length image foresight to future work.

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

## THE USE OF LARGE LANGUAGE MODELS

We only use LLMs as a tool for improving the quality of writing. We manually write the complete version of the paper. The output of the LLMs is merely used as a synonym replacement for some parts of our manually-written version. LLMs are not used for any creative work such as the research ideation or the design of experiments.

ETHICS STATEMENT

In this work, we are adhere to the code of ethics. This work does not involve human subjects, personal data, or sensitive information. All training data are synthesized by publicly available environment simulator. Our MBRL approach is task-agnostic, introducing no prior biases. We advocate for thorough testing and safety evaluations before deploying this reinforcement learning system in broader applications, especially physical systems.

REPRODUCIBILITY STATEMENT

We submit the source code as part of supplementary materials. Following the experiment setup in Table 1, all the results can be reproduced on publicly available RL environments and open source code repositories.

# A  MODEL DETAILS

Table 1: ResDreamer and DreamerV3 baseline model sizes

| **Configurations** | DreamerV3 | ResDreamer (50Mx2) | ResDreamer (100Mx2) |
|---|---|---|---|
| Foresight horizon | 0 | 4 | 4 |
| Recurrent $h_t$ size | 6144 | 4096 | 6144 |
| Recurrent $z_t$ size | $32 \times 48$ | $32 \times 32$ | $32 \times 48$ |
| Hidden size | 768 | 512 | 768 |
| Encoder CNN channels | 48 | 32 | 48 |
| Decoder CNN channels | 32 | 32 | 32 |
| hierarchies | 1 | 2 | 2 |
| Total parameters | 109.5M | 92.0M | 192.7M |
| Total training hours | 6.2 | 12.3 | 14.5 |

Figure 2 intuitively illustrates the data flow diagram of open-loop imagination and how it constructs enhanced visual observations during training. Our hierarchical model extends the process of updating the internal recurrent state based on observations. See Algorithm 1 for details.

---

**Algorithm 1** Update the recurrent state of ResDreamer upon observation

---

**Input:** recurrent state $s_t$, raw observation $o_{\text{raw}}$.
**Output:** recurrent state $s_{t+1}$, world model losses $\mathcal{L}_{\text{dyn}}(\phi), \mathcal{L}_{\text{rep}}(\phi), \mathcal{L}_{\text{rec}}(\phi)$.
 1: Open-loop rollout imaginary state-action trajectory $\left\{ \hat{s}^{0:L-1}, a \right\}_{t+1:t+H}$
 2: initiate $o_{\text{res}}^k$ with empty set.
 3: **for** each $k = 0, 1, \cdots, L-1$ **do**
 4:     Compute $o_{\text{imag}}^k$ with Eq. 3.
 5:     Compute $o_t^k$ with Eq. 4.
 6:     $z_t^k \leftarrow \text{sample}\left[ q_\phi \left( z_t^k \mid h_t^k, o_t^k \right) \right].$                                   ▷ Encoder
 7:     $\hat{z}_t^k \leftarrow \text{sample}\left[ p_\phi \left( z_t^k \mid h_t^k \right) \right].$                                   ▷ Predictor
 8:     Compute prediction loss $\mathcal{L}_{\text{dyn}}^k(\phi)$ and representation loss $\mathcal{L}_{\text{rep}}^k(\phi)$ with Eq. 7.
 9:     $h_{t+1}^k \leftarrow S_\phi \left( z_t^k, h_t^k, a_t^k \right).$                                   ▷ Sequence model
10:     Compute sensory signal reconstruction $\hat{o}_t^k = \left\{ \hat{o}_{\text{raw}}^k, \hat{o}_{\text{res}}^k \right\}_t.$                                   ▷ Decoder
11:     Compute reconstruction loss $\mathcal{L}_{\text{rec}}^k(\phi)$ with Eq. 6.
12:     Compute $o_{\text{res}}^k$ with Eq. 2.
13: **end for**
14: **return** $s_{t+1}, \mathcal{L}_{\text{dyn}}(\phi), \mathcal{L}_{\text{rep}}(\phi), \mathcal{L}_{\text{rec}}(\phi).$

---

The sequence of environmental interactions stored in the replay buffer is utilized only for training the representational learning of the world model, while policy improvement relies exclusively

on imagined trajectories. Consequently, the training pipeline and the environment interaction are entirely asynchronous. For a detailed description of the training pipeline, refer to Algorithm 2.

---

**Algorithm 2** The training pipeline of ResDreamer

---
1: initiate parameters $\phi, \theta, \psi$.
2: initiate carried state $s_{\text{carry}}$.
3: **while** not converged **do**
4:                                                         ▷ World model representation learning
5:     Sample a environmental interaction sequence $\{o_{\text{raw}}, a\}_{0:T-1}$ from replay buffer.
6:     **for** each $t = 0, 1, \cdots, T-1$ **do**
7:         Update the $s_{\text{carry}}$ upon $\{o_{\text{raw}}\}_t$ with Algorithm 1.
8:         Store trajectory feature $\{h_t^{0:L-1}, z_t^{0:L-1}\}$ and losses $\mathcal{L}_{\text{dyn}}(\phi), \mathcal{L}_{\text{rep}}(\phi), \mathcal{L}_{\text{rec}}(\phi)$.
9:     **end for**
10:                                                               ▷ Actor-critic learning
11:     Stack feature sequence $F \leftarrow \{h_{0:T-1}^{0:L-1}, z_{0:T-1}^{0:L-1}\}$.
12:     Compute the bootstrapped $\lambda$-return $R_t^{\lambda}$ and critic loss $\mathcal{L}(\theta)$ with Eq. 9.
13:     View $F$ as a batch of entry points sized $T$.
14:     Open-loop rollout imaginary state-action trajectory of $B$ time-steps starting at entry points batch $F$.
15:     **for** each imaginary trajectory $\{\hat{s}_{0:B-1}, a_{0:B-1}\}$ **do**
16:         Compute the normalized return and actor loss $\mathcal{L}(\psi)$ with Eq. 10.
17:     **end for**
18:     Back propagate losses $\mathcal{L}_{\text{dyn}}(\phi), \mathcal{L}_{\text{rep}}(\phi), \mathcal{L}_{\text{rec}}(\phi), \mathcal{L}(\theta), \mathcal{L}(\psi)$.
19:     Optimize parameters $\phi, \theta, \psi$.
20: **end while**

---

## B  ENVIRONMENT DETAILS

MineDojo agent's initial inventory includes a iron sword, shield, and a full suite of iron armors across all tasks. The maximum number of time-steps for one episode is 1000. For other specifications, see Table 2.

Table 2: MineDojo tasks specifications.

| Mobs | Biome | Mob Features | MineClip prompt |
|------|-------|--------------|-----------------|
| Spider | extreme hills | Fast movement | combat a spider in night extreme hills with a iron sword, shield, and a full suite of iron armors |
| Shulker | end | Shoots guided bullets which causes floating | combat a shulker in the end with a iron sword, shield, and a full suite of iron armors |
| Wolf | taiga | More agile, group attacks | combat a wolf in taiga with a iron sword, shield, and a full suite of iron armors |
| Skeleton | extreme hills | Accurate ranged attacks with arrows | combat a skeleton in night extreme hills with a iron sword, shield, and a full suite of iron armors |
| Ghast | nether | Flying, ranged attacks with explosive fireball, terrain destruction | combat a ghast in nether with a iron sword, shield, and a full suite of iron armors |

As shown in Table 2, the five Mobs each possess distinct characteristics. Each episode terminates upon timeout or when the agent's health reaches zero, which implies that the agent must not only explore and approach enemies but also learn to evade attacks or defend with a shield. The rich interaction mechanisms thoroughly test the generalization capabilities of RL algorithms.

Table 3: Ablation result

| Configurations | hierarchy | Rollout Hint | Residual Connection | Success Rate |
|---|---|---|---|---|
| ResDreamer (50Mx3) | 3 | ✓ | ✓ | 0.776 |
| ResDreamer (50Mx2) | 2 | ✓ | ✓ | 0.727 |
| ResDreamer (Only residual hints) | 2 | | ✓ | 0.563 |
| Dreamer (With roll-out foresight) | 1 | ✓ | | 0.559 |
| ResDreamer (Only rollout hints) | 2 | ✓ | | 0.400 |
| ResDreamer (Heads conditioned on all) | 2 | ✓ | ✓ | 0.377 |

## C  ADDITIONAL EXPERIMENTS

In order to rule out factors ofinsufficient training and to demonstrate a more complete successful dynamic, we additionally train ResDreamer (50M×2) as well as a enhanced Dreamer V3 baseline for 2.5M environment steps. The single-layer DreamerV3 is enhanced with the same online computational image foresight, but without hierarchical layers or residual modulation.

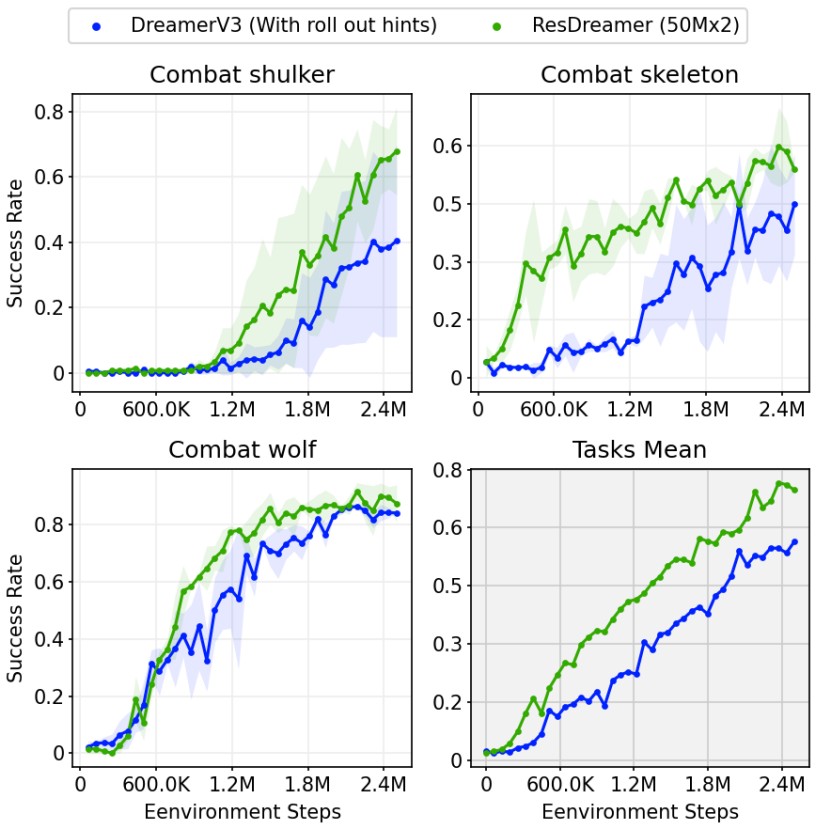

Figure 8: Comparison of ResDreamer and Dreamer V3 in 2.5M steps setup.

Even with significantly longer training, ResDreamer continues to improve and substantially outperforms the enhanced DreamerV3 baseline. This further strengthens our original conclusions.

## D  BASELINE INTRODUCTION

### D.1  SELECTED METHODS

We compare ResDreamer with strong Minecraft RL algorithms, including:

DreamerV3 (Hafner et al., 2025): A model-based RL foundation model. DreamerV3 is trained from scratch without demonstrations and domain knowledge. It generates future latent states recurrently with a non-hierarchical world model.

STEVE-1 (Lifshitz et al., 2023): An finetuned Video Pretraining (VPT) model for open-ended text and visual instructions following. It is post trained through self-supervised behavioral cloning. We test its zero-shoot text instructions following performance in MineDojo tasks.

PTGM (Yuan et al., 2024): A hierarchical approach integrating a high-level task goal generation strategy and a low-level goal-conditioned RL strategy. The high-level goal strategy is pretrained on large-scale, task-agnostic datasets, while the low-level strategy is learned online through RL. We utilize the open-source upper-layer strategy parameters of PTGM and evaluate its online training performance on MineDojo tasks using the default configuration of PTGM code-base.

### D.2  UNSELECTED METHODS

We provide introductions of other strong Minecraft agents and the reasons we do not compare ResDreamer with them.

LS-Imagine (Li et al., 2024): An MBRL method that achieves arbitrary time-span reasoning through dual-branch prediction. It is based on DreamerV3, but it supports long-term prediction by simulating jumping to the vicinity of navigation targets through cropping observation. However, combat missions are different from navigation and exploration. Factors such as terrain, enemy reactions, etc. have a significant impact on the expected return, and cutting the images disrupts the data distribution. For instance, it is not reasonable to jump to flying enemies like ghasts by cropping the image.

Voyager (Wang et al., 2023a), JARVIS-1 (Wang et al., 2023b), MC-Planner (Wang et al., 2023c), RL-GPT (Liu et al., 2024): Open-Ended embodied agents that integrates RL with LLM. They adopt heterogeneous hierarchical models, leveraging the prior knowledge of LLMs to achieve task decomposition, long-term planning, code as strategy, and lifelong skill accumulation. Their focus lies in the integration and interaction methods between LLMs and RL, emphasizing the evaluation of an agent's efficiency in accumulating atomic skills and activating technological milestones. Our proposed ResDreamer is a model-based RL foundation model, focusing on evaluating the data efficiency, scalability, and interpretability. ResDreamer can work together with all kinds of upper layer LLMs as a more powerful RL algorithm.

ROCKET-2 (Cai et al., 2025a), ROCKET-3 (Cai et al., 2025b) SkillDiscovery (Deng et al., 2025), JarvisVLA (Li et al., 2025): Open-world VLA agents powered by imitation learning (IL) and prior knowledge of visual foundation model such as SAM (Kirillov et al., 2023). VLA agents focus on following open instructions within a broader range of atomic skills and their combinations. However, ResDreamer is a MBRL foundation model trained without any prior knowledge. ResDreamer focuses on developing a task-agnostic and domain general hierarchical world model method.

## E  ADDITIONAL VISUALIZATION

Residual Enhanced Visual Observation is the main innovation of this work. To visually demonstrate the structure of this visual foresight and the planning information it provides, we visualize the observation sequence of the agent during its combat with the wolf in Figure 9.

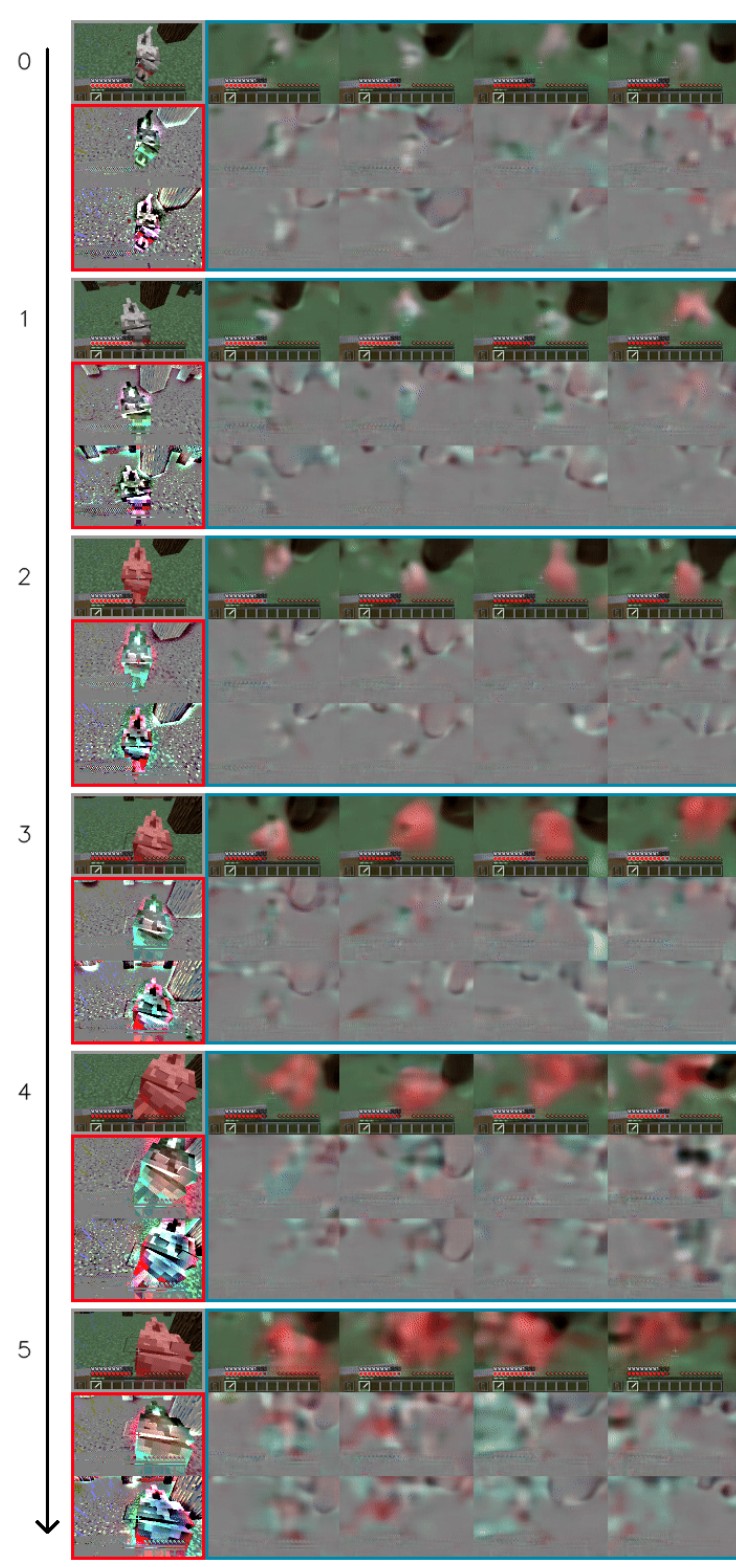

Figure 9: Visualization of all the observations of a ResDreamer with three hierarchies. **Gray**: raw observation. **Red**: residual observation. **Blue**: original open-loop imagination.

