# OpenReview forum: "Envision the Future in Open-World Dynamic Tasks by a Hierarchical World Model with Residual Enhanced Foresight"
_ICLR.cc/2026/Conference — Submitted to ICLR 2026_

### Official Review · Reviewer_q9v4 · 2025-10-29

**Soundness:** 2
**Presentation:** 2
**Contribution:** 2
**Rating:** 4
**Confidence:** 4

**Summary:**

This paper proposes a hierarchical world model, ResDreamer, for dynamic tasks in the open world, such as battle scenarios in Minecraft. The core idea is that, in a hierarchical structure, the high-level model constructs a more comprehensive representation of the world by observing the visual reconstruction residuals from the low-level model and feeding back the modified “vision foresight” to it, thereby enhancing planning and decision-making capabilities. This method does not rely on the language model or external pre-training knowledge; it relies solely on self-supervised interactive learning, which offers good sampling efficiency, parameter efficiency, and scalability.

**Strengths:**

- Combining the idea of "predictive coding” in neuroscience with hierarchical Reinforcement Learning, the mechanism of using reconstruction residuals as inter-layer communication signals is proposed, which is relatively novel in the existing MBRL literature.
- ResDreamer's information flow design (high-level modeling, low-level errors, low-level receiving high-level corrected visual cues) mimics the prediction error processing mechanisms of the cerebral cortex (e.g., Rao & Ballard, 1999), not only improving model interpretability, but also improving the accuracy of prediction processing. It also provides a new idea for future brain-like RL.
- The results on five high-difficulty combat missions of MineDojo show that ResDreamer (especially the 100M × 2 configuration) outperforms the baseline in both sample efficiency and success rate, and even exhibits the only feasible solution ability in extremely difficult tasks such as confronting Shulkers.

**Weaknesses:**

- Although Figure 5 contains variants such as “only residual hints,” a key control group is missing: a hierarchical Dreamer who completely removes residual signals and uses only the original image + imagined trajectories. This makes it difficult to determine whether the performance improvement comes from the hierarchy itself or from the specific mechanism of residual enhancement.
- All experiments were conducted in Minedojo's combat subtasks with explicit friend-foe relationships and sparse reward structures. But the challenges of open-world RL also include long-term exploration, tool use, and multi-agent collaboration. ResDreamer's claim to be a “Universal World Model” is undermined by its lack of validation for navigation, construction, or hybrid missions.
- The paper acknowledges that fixed field length is the main limitation (Section 5), but the effect of different H (e.g., H = 4 vs H = 8) on performance is not explored in experiments. If H is too small, it will be invalid. If H is too large, it will be expensive and fuzzy. There is no sensitivity analysis or dynamic adjustment strategy for H, which undermines the method's practicability.

**Questions:**

- Can you provide a hierarchical Dreamer baseline without residual connections (i. e. only hierarchical but no residual communication)?
This will directly verify whether residual enhancement is a core contribution rather than a benefit of simple stratification.
- Is ResDreamer suitable for non-combat open missions? If the author has a preliminary experiment, even if it does not reach SOTA, please briefly describe it to support its broader applicability.
- The paper mentions "the distribution of lower residual observations shifts during the training process of lower layer models” (Section 4.2), but does not specify whether stabilization strategies such as gradient stopping or curriculum learning are employed. Please clarify the details of the training dynamics.

---

> ### Author Response · Authors · 2025-11-23
> **Response to Reviewer q9v4 (Part 1)**
>
> Dear Reviewer q9v4,
>
> We thank you for your thorough review and constructive suggestions. We have added the supplementary experiments you mentioned. We hope that our response will clarify your query and address most of your concerns.
>
> ## Additional Analytical Results
>
> > W1 (Q1):  A key control group is missing: a hierarchical Dreamer who completely removes residual signals and uses only the original image + imagined trajectories.
>
> We strongly agree that ablation experiments on residual correction are necessary. As suggested, we have conducted the requested ablation studies. Ablation results are summarized in the table below
>
> |#|Configuration|Hierarchy|Rollout Hint|Residual Connection|Actor/Critic Input|Success Rate|
> |-|-|-|-|-|-|-|
> |1|ResDreamer (50M×3)|3 layers|✓|✓|Bottom layer only|0.776|
> |2|ResDreamer (50M×2) – main model|2 layers|✓|✓|Bottom layer only|0.727|
> |3|ResDreamer (Residual-only, no hint)|2 layers||✓|Bottom layer only|0.563|
> |4  |DreamerV3 (50M) + image foresight (no hierarchy)|1 layer|✓||Single layer|0.559|
> |5|ResDreamer (Hint-only, no residual)|2 layers|✓||Concatenated latents from all layers|0.400|
> |6|ResDreamer (Heads conditioned on all)|2 layers|✓|✓|Concatenated latents from all layers|0.377  |
>
>
> Specifically, the configuration 4 & 5 both show removing the residual causes performance drops.
>
> Our added ablations, together with the original ones, have verified that the combination of hierarchical structure, hint rollout, and residual modulation is significantly more effective than any subset of these components.
>
> ## Additional Benchmark and Other Open-World RL Challenges
>
> > W2:  All experiments were conducted in Minedojo's combat subtasks with explicit friend-foe relationships and sparse reward structures. But the challenges of open-world RL also include long-term exploration, tool use, and multi-agent collaboration. ResDreamer's claim to be a "Universal World Model" is undermined by its lack of validation for navigation, construction, or hybrid missions.
>
> Thanks for your insightful comment. ResDreamer is designed to be domain-general, as its residual communication mechanism is fully self-supervised and requires no domain-specific priors (e.g., offline data or pre-trained models). Long-horizon open-world challenges (e.g., lifelong skill discovery, tool use, and multi-agent collaboration) are typically addressed by hybrid methods such as Voyager [1], JARVIS-1 [2], and RL-GPT [3], which heavily rely on pre-trained LLMs and offline data.
>
> Our work primarily focuses on the contribution of the proposed hierarchical world model and residual-modulated visual reasoning in highly dynamic interaction tasks. While we have not yet evaluated on other challenges such as tool use and construction, the strong generalization of DreamerV3 combined with our foresight reasoning provides a promising foundation for future exploration.
>
> > Q2:  Is ResDreamer suitable for non-combat open missions? If the author has a preliminary experiment, even if it does not reach SOTA, please briefly describe it to support its broader applicability.
>
> Yes. We have tested ResDreamer on 5 most difficult DMC Vision [4] tasks that Dreamer V3 did not converge to full score. ResDreamer consistently outperforms Dreamer V3 with aligned experiment setup.
> Score curves: [https://anonymous.4open.science/r/ResDreamer-F33C/img/curves_score_dmc_vision.png](https://anonymous.4open.science/r/ResDreamer-F33C/img/curves_score_dmc_vision.png)

---

> ### Author Response · Authors · 2025-11-23
> **Response to Reviewer q9v4 (Part 2)**
>
> ## Sensitivity Analysis on Foresight Horizon
>
> > W3: The effect of different H (e.g., H = 4 vs H = 8) on performance is not explored in experiments. If H is too small, it will be invalid. If H is too large, it will be expensive and fuzzy. There is no sensitivity analysis or dynamic adjustment strategy for H, which undermines the method's practicability.
>
> Thanks for the very insightful and constructive suggestion. We followed this recommendation and added sensitivity analysis experiments for Foresight Horizon. We evaluate different rollout horizons (H=4, 8, 16) on the continuous visual control benchmark DMC Vision [4]. To keep the total number of imagined frames fixed at 4 (same computational budget), we vary the stride D = 1 (H=4), D=2 (H=8), and D=4 (H=16).
> Score curves: [https://anonymous.4open.science/r/ResDreamer-F33C/img/curves_score_dmc_vision.png](https://anonymous.4open.science/r/ResDreamer-F33C/img/curves_score_dmc_vision.png)
>
> Generally, after sufficient training, foresight with longer horizon and larger stride is more informative, but the convergence speed may be slower in some cases. Here are two specific and interesting findings:
>
> - In "DMC Reacher Hard" which requests  precise control of a two-link robotic arm, all ResDreamer configurations converge to the full score within 1M steps. The reasoning table with a shorter time window converges the fastest.
> - The "DMC Hopper Hop" requires a 4-joint legged robot to perform a high-speed forward leap while maintaining balance. The agent with rollout horizon H=16 significantly outperforms other setup for 16 steps roughly covers the entire cycle of a single jump.
>
> ## Stable Training via Batch-Normalized Residual Image
>
> Since the upper-level world model must learn and reconstruct residual images, we need to minimize their distribution shift during training. To this end, we apply batch normalization to the residual images, with running mean and variance updated via exponential moving average (EMA) using a rate of 0.01.
>
> Although batch normalization inevitably shifts the distribution of reconstructed residuals away from the original, precise pixel-level fidelity with future frames is not required. The residual-corrected image foresight primarily needs to be maximally informative rather than strictly distribution-matched.
>
> To isolate the contribution of our residual reasoning representation as cleanly as possible, we did not employ any additional training strategies (e.g., adaptive curriculum learning).
>
> ---
>
> We hope these revisions and additional experiments adequately address your concerns. We are happy to provide additional clarifications, experiments, or discussions upon request.
>
> ## *References*
>
> [1] Wang, G., Xie, Y., Jiang, Y., Mandlekar, A., Xiao, C., Zhu, Y., ... & Anandkumar, A. (2023). Voyager: An open-ended embodied agent with large language models. arXiv preprint arXiv:2305.16291.
>
> [2] Wang, Z., Cai, S., Liu, A., Jin, Y., Hou, J., Zhang, B., ... & Liang, Y. (2024). Jarvis-1: Open-world multi-task agents with memory-augmented multimodal language models. *IEEE Transactions on Pattern Analysis and Machine Intelligence*.
>
> [3] Liu, S., Yuan, H., Hu, M., Li, Y., Chen, Y., Liu, S., ... & Jia, J. (2024). Rl-gpt: Integrating reinforcement learning and code-as-policy. *Advances in Neural Information Processing Systems*, *37*, 28430-28459.
>
> [4] Ortiz, J., Dedieu, A., Lehrach, W., Guntupalli, J. S., Wendelken, C., Humayun, A., ... & Murphy, K. P. (2024). DMC-VB: A Benchmark for Representation Learning for Control with Visual Distractors. *Advances in Neural Information Processing Systems*, *37*, 6574-6602.

---

> > ### Comment · Reviewer_q9v4 · 2025-11-24
> >
> > I thank the authors for their response. My concerns have been largely addressed, except for one regarding the discussion of ResDreamer as a "general world model." While the additional DMC experiments are appreciated, both MineDojo and DMC remain highly simplified simulation environments. These settings lack the complexity of the real physical world—such as rich physical interactions, diverse material textures, and realistic hardware latency and sensor noise. As a result, ResDreamer's strong performance in these environments may not yet be sufficient to support its viability as a general world model for real-world applications such as robotic manipulation or other embodied AI tasks.
> >
> > To further strengthen the method's practical relevance, it would be valuable to evaluate ResDreamer on more realistic benchmarks for robotic manipulation, such as SimpleEnv [1], CALVIN [2], or BridgeData V2 [3]. Such validation could significantly enhance the persuasiveness of the method's utility and generalizability.
> >
> > [1] Evaluating Real-World Robot Manipulation Policies in Simulation CoRL 2024
> >
> > [2] CALVIN - A benchmark for Language-Conditioned Policy Learning for Long-Horizon Robot Manipulation Tasks R-AL 2022
> >
> > [3] BridgeData V2: A Dataset for Robot Learning at Scale CoRL 2023

---

> > > ### Author Response · Authors · 2025-11-26
> > > **Response to Reviewer q9v4**
> > >
> > > ## ResDreamer is Reasoning Method Free of Domain Knowledge
> > >
> > > Thank you for further explaining your reasonable concerns. We have replaced the "domain-general" throughout the manuscript with the following clearer phrasing:
> > >
> > > > We propose a hierarchical world model that requires no domain-specific knowledge.
> > >
> > > We consider the realistic benchmarks you provided for robotics to be of great value. We plan on building language-conditioned ResDreamer in our future work.
> > >
> > > Regarding the value of our approach for real-world physical AI, ResDreamer provides a brain-inspired sensory-signal reasoning mechanism that complements prior-knowledge-based approaches.
> > >
> > > Once again, thank you for your insightful and constructive comments. Please do not hesitate to let us know if any concerns remain.

---

### Official Review · Reviewer_eFj2 · 2025-10-31

**Soundness:** 2
**Presentation:** 2
**Contribution:** 2
**Rating:** 4
**Confidence:** 3

**Summary:**

Reinforcement learning algorithms have become increasingly important in the last years. One of the major challenges to deep reinforcement learning algorithms is sample efficiency, especially for long-term tasks. To address this issue an approach is learn a world model to train an agent entirely in imagination, eliminating the need for direct environment interaction during training. To improve exploration capabilities and sample efficiency the authors propose a hierarchical world model approach (ResDreamer) based on visual planning representations. Numerical experiments are performed (on 5 combat tasks in MineDojo) and results are compared DreamerV3 as a baseline.

**Strengths:**

S1 The considered problem is interesting and relevant.

S2 The paper is overall well-written.

**Weaknesses:**

W1 The paper does not provide analytical results.

W2 The evaluation is too small, contains only few handpicked(?) examples and is hence not fully convincing. Further, DreamerV3 is not clearly outperformed.

W3 Comparisons against other strong baselines beyond DreamerV3 are missing.

**Questions:**

Besides DreamerV3 you should also compare your results against further baselines, such as IRIS (Micheli et al., 2023), TWM (Robine et al., 2023) and Hieros (Mattes et al., 2024), which also uses a hierachical approach.

What are the required computational costs? Does the approach scale to larger problems? In which type of tasks the hierarchical structure particularly pays off?

Which hyper parameters are required? How are they automatically chosen/tuned?

---

> ### Author Response · Authors · 2025-11-23
> **Response to Reviewer eFj2 (Part 1)**
>
> Dear Reviewer eFj2,
>
> We thank you for your thorough and insightful review. We hope that our response will clarify your query and address most of your concerns.
>
> ## Summary of analytical results
>
> > W1: The paper does not provide analytical results.
>
> As you mentioned, we have added 3 additional analysis experiments. Please refer to the global comment **Additional Experiment Summary** for setup details. Ablation results are summarized in the table below
>
> |#|Configuration|Hierarchy|Rollout Hint|Residual Connection|Actor/Critic Input|Success Rate|
> |-|-|-|-|-|-|-|
> |1|ResDreamer (50M×3)|3 layers|✓|✓|Bottom layer only|0.776|
> |2|ResDreamer (50M×2) – main model|2 layers|✓|✓|Bottom layer only|0.727|
> |3|ResDreamer (Residual-only, no hint)|2 layers||✓|Bottom layer only|0.563|
> |4|DreamerV3 (50M) + image foresight (no hierarchy)|1 layer|✓||Single layer|0.559|
> |5|ResDreamer (Hint-only, no residual)|2 layers|✓||Concatenated latents from all layers|0.400|
> |6|ResDreamer (Heads conditioned on all)|2 layers|✓|✓|Concatenated latents from all layers|0.377|
>
>
> - Config 1 & 2 proves the scalibility of ResDreamer.
> - Config 3 & 4 show removing either the hint or the residual causes massive drops. The residual alone is insufficient, and the hint alone is not as helpful.
> - Config 5 & 6 demonstrate conditioning actor-critic on concatenated latents from all layers leads to poor performance. This confirms that before the world model fully converges, unstable high-level latent representations can slow down RL convergence. In contrast, residual-modulated image foresight provides relatively stable reasoning representations.
>
> ## Task Benchmark
>
> > W2 The evaluation is too small, contains only few handpicked(?) examples and is hence not fully convincing
>
> We focus on 3D open world environments with dynamic interactive objects. In Minedojo benchmark, we selected rich environment styles and combat mechanisms to verify the mission generalization of the model.
>
> |Mobs|Biome|Mob Features|
> |-|-|-|
> |Spider|extreme hills|Fast movement|
> |Shulker|end|Shoots guided bullets which causes floating|
> |Wolf|taiga|More agile, group attacks|
> |Skeleton|extreme hills|Accurate ranged attacks with arrows|
> |Ghast|nether|Flying, ranged attacks with explosive fireball, terrain destruction|
>
>
> The five tasks vary in difficulty to demonstrate the dynamics of success at different convergence stages.
>
> We completely agree that evaluation on expanded benchmarks makes our conclusion more convincing.
>
> In our added experiment, we compare ResDreamer with different foresight horizon (H=4, 8, 16) on DeepMind Control Vision continuous control suite. We select the 5 most difficult DMC
> Vision tasks that Dreamer V3 did not converge to full score.
>
> Score curves: [https://anonymous.4open.science/r/ResDreamer-F33C/img/curves_score_dmc_vision.png](https://anonymous.4open.science/r/ResDreamer-F33C/img/curves_score_dmc_vision.png)
>
> ResDreamer continues to outperform the DreamerV3 baseline in aligned configurations. This further strengthens our original conclusions.
>
> ## Significance of Improvement
>
> > W2: DreamerV3 is not clearly outperformed.
>
> - **Consistent and Stable Improvements**: ResDreamer shows stable improvements in cross-domain tasks of varying difficulty with unified hyperparameters, including Minedojo combat and DMC visual control
> - **Notable Improvements in some tasks**: ResDreamer exhibits the only feasible solution ability in extremely difficult tasks such as confronting shulkers. All ResDreamer configurations converged to the full score on the "Reacher hard" task within 1M steps, but DreamerV3 did not achieve this. ResDreamer (H=16)  task score is about 2 times higher than DreamerV3 on "Hopper hop".

---

> ### Author Response · Authors · 2025-11-23
> **Response to Reviewer eFj2 (Part 2)**
>
> ## Additional Baseline
>
> > W3(Q1): comparisons against other strong baselines, such as IRIS [1], TWM [2] and Hieros [3].
>
> Thanks again for the constructive suggestion. We have reproduced IRIS on Minedojo and run a experiment using three random seeds and the official default hyperparameters. The only necessary changes were:
>
> 1. Adding support for MineDojo's MultiDiscrete action space (instead of the original flat Discrete used on Atari).
> 2. Setting MineDojo observations to 64×64 (instead of our usual 80×128) to match IRIS's hard-coded input resolution. This might weaken the model's capabilities, but is required for compatibility.
>
> We have trained IRIS under default configuration for 7 days. So far, IRIS has failed to achieve meaningful success even on the easiest Combat Spider task throughout 500K environment steps.
>
> Results (three seeds): [https://anonymous.4open.science/r/IRIS-Minedojo-2CB0/results/curves_iris.png](https://anonymous.4open.science/r/IRIS-Minedojo-2CB0/results/curves_iris.png)
>
> Our current conclusion regarding the reproduction of IRIS on Minedojo is that the default configuration is unable to complete the Minedojo task. Further hyperparameter tuning is still required.
>
> To the best of our knowledge, we are the first to test domain-general reinforcement learning methods based on hierarchical world models in a 3D open-world game environment. IRIS [1], TWM [2] and and Hieros [3] is only officially tested on 2D Atari environment.
>
> We plan on doing further baseline reproduction and tuning in future work. Further explanation of our baseline choices is included in Appendix D "Baseline Introduction".
>
> ## Computational Cost and When ResDreamer More Significantly Pays Off
>
> ### Compute budget
>
> > Q2: What are the required computational costs? Does the approach scale to larger problems?
>
> We aligned parameter and train ratio (optimizer updates per environment step) to enable a fair comparison between ResDreamer and Dreamer V3 under the same computational budget. The additional total wall time of ResDreamer is derived from the foresight image obtained through visual grounded reasoning.
>
> |**Compute budget in main experiment**  |ResDreamer (50Mx2) |ResDreamer (100Mx2)|Dreamer V3 (100M)|
> |-|-|-|-|
> |Total parameters|92.0M|192.7M|109.5M|
> |Total training hours for (1M steps)|12.3|14.5|6.2|
>
>
> ResDreamer incurs ~2× wall-time overhead but achieves higher sample efficiency.
>
> Despite this, thanks to the efficient JAX implementation, ResDreamer has significantly better computational efficiency than other model based RL method such as IRIS (~14 days in default configuration for 1M environment steps). We plan to apply ResDreamer to larger problems such as robot manipulation and autonomous driving in future work.
>
> ### When ResDreamer More Significantly Pays Off
>
> > Q2: In which type of tasks the hierarchical structure particularly pays off?
>
> 1. ResDreamer significantly pays off when task-related objects change in complex dynamic mechanisms
>
>     ResDreamer (100Mx2) is the only method that solves combating a shulker in $1\times 10^6$ environment steps. The Shulker launches a guided projectile that causes prolonged levitation and causes fall damage afterwards. This poses complex and challenging dynamic interaction mechanisms for the agent.
>
>     We found that the residual-modulated visual reasoning representation provided significant assistance in understanding this dynamic interaction mechanism. On the one hand, the feedforward channel provides modulated foresight image which makes the observation source move informative. On the other hand, the feedback channel allows error propagation upward, resulting in a more comprehensive inner world representation.
> 2. ResDreamer significantly pays off when the foresight horizon matches the temporal extent of the task-relevant dynamics.
>
>     DMC Hopper Hop requires a 4-joint legged robot to perform a high-speed forward leap while maintaining balance. The agent with rollout horizon H=16 significantly outperforms other setup for 16 steps roughly covers the entire cycle of a single jump.

---

> > ### Author Response · Authors · 2025-11-24
> > **Response to Reviewer eFj2 (Part 3)**
> >
> > ## Hyperparameters
> >
> > > Which hyper parameters are required? How are they automatically chosen/tuned?
> >
> > Thank you for the very precise questions. We follow the official DreamerV3 hyperparameters without any task-specific tuning.
> > Benefits from DreamerV3’s robust generalization capabilities, ResDreamer can be trained across diverse tasks without hyperparameter tuning.
> >
> > The only non-trivial hyperparameter introduced by ResDreamer is the rollout horizon H, we have added a systematic sensitivity study (H = 4, 8, 16) on the continuous-visual-control benchmark DMC Vision [8] while keeping the total number of imagined frames fixed at 4. Results are provided in the “Additional Experiment Summary” section and here: [https://anonymous.4open.science/r/ResDreamer-F33C/img/curves_score_dmc_vision.png](https://anonymous.4open.science/r/ResDreamer-F33C/img/curves_score_dmc_vision.png)
> >
> > ---
> >
> > We hope these revisions and additional experiments adequately address your concerns. We are happy to provide additional clarifications, experiments, or discussions upon request.
> >
> > ## *References*
> >
> > [1] Micheli, V., Alonso, E., & Fleuret, F. (2022). Transformers are sample-efficient world models. *arXiv preprint arXiv:2209.00588*.
> >
> > [2] Robine, J., Höftmann, M., Uelwer, T., & Harmeling, S. Transformer-based world models are happy with 100k interactions, 2023. *URL *[*https://arxiv*](https://arxiv)*. org/abs/2303.07109*.
> >
> > [3] Mattes, P., Schlosser, R., & Herbrich, R. (2023). Hieros: Hierarchical imagination on structured state space sequence world models. *arXiv preprint arXiv:2310.05167*.

---

### Official Review · Reviewer_asHi · 2025-11-03

**Soundness:** 2
**Presentation:** 1
**Contribution:** 3
**Rating:** 4
**Confidence:** 5

**Summary:**

ResDreamer is a hierarchical world model for RL that passes reconstruction residuals upward so higher layers learn what lower layers failed to explain and provides feed back corrections to improve lower-layer foresight. Each layer consumes an “enhanced observation” consisting of raw pixels, a residual stack, and an imaginary hint built from open-loop predicted frames plus residual corrections from the next layer, with no gradients flowing between layers. The world model optimizes reconstruction of raw and residual signals and uses paired KL terms to balance representation and dynamics prediction. Training rolls out imagined trajectories to supply hints and to learn the actor-critic largely from imagination. Experiments on five MineDojo combat tasks show success-rate curves over 1M environment steps. Compared to DreamerV3, PTGM, and STEVE-1, ResDreamer shows improvements in success with 50Mx2 and 100Mx2 parameters, and performance improves when stacking to three layers. Ablations indicate the residual-corrected foresight is important. The paper concludes that the design scales with roughly linear inter-layer bandwidth and yields better sample and parameter efficiency for open-world dynamic tasks.

**Strengths:**

1. The idea is sound. Fairly novel algorithmic improvement over Dreamer-v3
2. The paper has managed to convert an idea from neuroscience, “predictive coding” or error based learning into a practical implementation for world models which is impressive.
3. The method is evaluated on a non-trivial benchmark (MineDojo) built on Minecraft style environment.
4. The authors also present a relevant ablation by increasing the hierarchy to 3 levels which confirms the scalability of the method

**Weaknesses:**

1. My biggest issue with the paper is the poor presentation of the idea and the lack of clarity in communication. I found it fairly difficult to understand the idea despite having extensively dealt with Residual connections, Dreamer and Predictive Coding myself. My concerns are that:

    a. Presentation of figures and mathematical expressions are average at best: In Figure 3, what do the red, green and gray dots represent? Do the background colors (red, green) indicate something? In Figure 2, what does the video symbol represent? Is it observation inputs or open-loop predictions? What do the dotted red circle represent? Perhaps a legend for all figures could help understand them quickly? In Figure 3 why are there blue frames after the yellow frames? I have asked a few other questions below.

    b. The language & communication is not upto the mark for a top-tier conference. I found it generally difficult to parse the paper. For example, o_imag^k is shown in Equation (2) but only defined and discussed before Equation (2). Few grammatical errors: “This has enabled the world model to advance towards the scalable ”ResNet era”. “In the field of visual MBRL, transformer, diffusion model are also known as effective world models”. “They have become the key to our hierarchical world model, enabling it to scale up with linearly increasing communication bandwidth.

2. I have several questions and issues regarding the algorithm, design choices and experiments. I would like the authors to clarify them clearly:

    a. Residual stacking: You state both o_res^k and o_imag^k have shape (h, w, 3*H), but Eq. (3) defines o_res^k only at time t. Do you stack residuals over t+1:t+H as well, or is the stated shape incorrect? Specify exact tensor shapes per layer and per time step.

    b. Normalization operator: Precisely define Norm_k(). Which axes are used for mean/variance. How is the EMA used? Provide the exact formula/expression.

    c. For Eq. (4), please detail the policy and temperature used for open-loop rollouts, the action horizon H per layer, and whether hints are recomputed every step or cached. Confirm that residuals from the upper layer come from an online or target decoder.

    d. You say no gradients pass between layers. Specify the exact detach points for o_raw, o_res^k, and o_imag^k via either the equations or one of the figures.

    e. Compute budget: Please report the number of imaginations per environment step per layer, total wall-clock hours, updates, etc for each method so sample efficiency isn’t confounded by extra compute.

    f. How many random seeds were used per task and algorithm? How are these results aggregated, and how is the confidence interval computed? It seems to me like the confidence intervals are showing a deviation from running average instead of aggregated numbers from multiple seeds. If that is the case, please provide the results for multiple seeds (atleast 3) since RL algorithms are noisy and single seed experiments are unreliable.

    g. Why were the training runs in Figure 4 reported only upto 1M steps? Especially for Combat shulker, Combat wolf and Combat skeleton? It seems like these experiments could have been run for longer, for a more clear idea of the success dyanmics.

3. Why were transformer based world models like IRIS [1] or pre-trained visual encoder based models like Dino-WM [2] not compared? The comparison would help guage the position of ResDreamer over other methods.

4. “However, as far as we know, there is no MBRL method that naturally builds a hierarchical representation learning architecture based on the reconstruction residuals of sensory signals.” I believe there are several works that have attempted this kind of hierarchical representations [3][4][5][6].

**Questions:**

Asked above.

Conclusion: I was really rooting for the authors when I began reading the paper and found the ideas interesting. But I have so many questions regarding the mechanics, design choices and experiments that I strongly feel it is premature to accept it without carefully assessing the contributions correctly. I would encourage the authors to clarify them during the discussion phase.

*References*

*[1] Micheli, V., Alonso, E., & Fleuret, F. (2023). Transformers are sample-efficient world models (IRIS). In Proceedings of the Eleventh International Conference on Learning Representations (ICLR 2023)*

*[2] Zhou, G., Pan, H., LeCun, Y., & Pinto, L. (2024). DINO-WM: World Models on Pre-trained Visual Features enable Zero-shot Planning. arXiv preprint arXiv:2411.04983*

*[3] Rao, R. P. N., et. al. (2024). Active predictive coding: A unifying neural model for active perception, compositional learning, and hierarchical planning. Neural Computation.*

*[4] Mounir, R., Vijayaraghavan, S., & Sarkar, S. (2023). STREAMER: Streaming representation learning and event segmentation in a hierarchical manner. In Advances in Neural Information Processing Systems 36 (NeurIPS 2023)*

*[5] Hansen, N., et al. (2025). Hierarchical world models as visual whole-body humanoid controllers. arXiv preprint arXiv:2501.01234*

*[6] Mattes, J., et al. (2023). Hieros: Hierarchical imagination on structured state-space sequence world models*

---

> ### Author Response · Authors · 2025-11-22
> **Response to Reviewer asHi (Part 1)**
>
> Dear Reviewer asHi,
>
> We are extremely grateful for your great efforts in reviewing our paper. We sincerely apologize for the confusion and extra time caused by the unclear presentation. In addition to clarifying and improving the paper presentation, we have added the supplementary experiments you mentioned.
>
> All results are reported as mean ± standard deviation over random seeds 0,1,2,3. Our DreamerV3 baseline uses the same hyperparameters and training schedule as the official DreamerV3 implementation to ensure a fair comparison. Other baselines use the default configuration of the official implementation.
>
> We hope that our response will clarify your query and address most of your concerns.
>
> ## Clarification on elements in figures and modification of presentation.
>
> Thanks for pointing out the unclear part in the presentation. We have added legends for all mentioned figures and carefully polished the language of the paper.
>
> > W1.a.Q1: In Figure 1, 2, what do the red, green and gray dots represent?
>
> Red, green and gray dots represent lower residual observation $o_{\text{res}}^k$, imaginary hint observation $o_{\text{imag}}^k$and raw environment observation $o_{\text{raw}}$ (Legends have been added).
>
> > W1.a.Q2: Do the background colors (red, green) indicate something?
>
> Yes. The colored backgrounds are visual cues highlighting which module processes which observation component (residual vs. hint vs. raw). Clear legends are now provided in both Figures 1 and 2.
>
> > W1.a.Q3: In Figure 2, what does the video symbol represent?
>
> It denotes image foresight generated via open-loop rollout of the world model. (Now explicitly explained in the legend.)
>
> > W1.a.Q4: What do the dotted red circle represent?
>
> They indicate the reconstructed estimate of the corresponding colored observation component (residual, hint, or raw) produced by the decoder.
>
> > W1.a.Q5: In Figure 3 why are there blue frames after the yellow frames?
>
> In Figure 3, both blue and yellow frames represent imaginary hint observation. We used yellow frames merely to distinguish foresight at the same time of next roll (the time stride of each row is 2). Comparing it with the next raw observation can easily verify that the foresight carries valid information.
>
> > W1.b: Problems in the language and mathematical expressions
>
> We sincerely appreciate your detailed feedback on writing and notation issues. We have rewritten the subsection "Visual Hint Structure and Residual Modeling", rearranged some equations and unified all mathematical symbols. We have conducted another full round of language polishing across the entire paper. We believe these weaknesses have now been largely resolved.

---

> ### Author Response · Authors · 2025-11-22
> **Response to Reviewer asHi (Part 2)**
>
> ## Clarification on the tensor shapes associated with imaginary hint observation
>
> > W2.a: Residual stacking and shape of imaginary hint observation.
>
> We do stack residuals over H-steps frames. We have clearly specified the precise tensor shapes related to each layer's observations in the paper. Time index is also changed to t:t+H for consistency with Eq. 1. We have corrected all of them in  the revised manuscript.
>
> Following is a character-style data flow diagram that depicts the complete observation o_t^k calculation and tensor shape for the general layer k.
>
> ```text
> =============================================
> Higher layer (k+1) Predicted residual rollout
> =============================================
>             │
>             ▼
> ┌──────────────────────────────────────────┐
> │   \hat{o}_res^{k+1}_{t:t+H} (H,h,w,3)    │
> └──────────────────────────────────────────┘
>             │
>             │      =============================================
>             │       Current layer (k) Predicted residual rollout
>             │      =============================================
>             │                       │
>             │                       ▼
>             │       ┌──────────────────────────────────────┐
>             │       │   \hat{o}_res^k_{t:t+H} (H,h,w,3)    │
>             │       └──────────────────────────────────────┘
>             │                       │
>             └───────────┬───────────┘
>                         │
>                         │ + element-wise add
>                         ▼
>         ┌──────────────────────────────┐
>         │     o_imag^k (H,h,w,3)       │
>         └──────────────────────────────┘
>             │
>             │ rearrange to shape (h,w,3H)
>             │
>             │
>             │   ┌─────────────────────┐      ============================
>             │   │   o_res^k (h,w,3)   │ ◄────      Lower layer (k-1)
>             │   └─────────────────────┘      Sensory reconstruction error
>             │              │                 ============================
>             │              │
>             ▼              ▼                 ┌──────────────────┐
>         o_imag^k_t       o_res^k_t           │ o_raw_t (h,w,3)  │
>     (imaginary hint)   (lower residual)      └──────────────────┘
>             │               │                       │
>             └───────────────┬───────────────────────┘
>                             │ Concat along channel axis
>                             │ sg(·) stop-gradient
>                             ▼
>                     ┌──────────────┐
>                     │     o_t^k    │   Final observation
>                     │  (h,w,3H+6)  │
>                     └──────────────┘
>                             │
>                     To k-th PPB encoder  z_t^k ~ q(z_t^k | h_t^k, o_t^k)
> ```

---

> ### Author Response · Authors · 2025-11-22
> **Response to Reviewer asHi (Part 3)**
>
> ## Clarification on **residual image** Normalization
>
> > W2.b: Normalization in **lower residual observation**
>
> Thank you for the thoughtful question. We believe that the following pseudo-code will clearly describe the normalization process of the residual image in our implementation.
>
> ```Python
> class NormalizeMeanStd:
>     def __init__(self, rate=0.01, limit=1e-8):
>         self.rate   = rate          # EMA update rate (β = 1 - rate)
>         self.limit  = limit         # minimum allowed std
>
>         self.ema_mean = 0.0         # E[x]
>         self.ema_sqrs = 0.0         # E[x²]
>
>     def update(self, batch):
>         """
>         Update statistics with a new batch of data.
>         """
>         batch_mean = batch.mean()          # scalar mean over the whole batch
>         batch_sqrs = (batch**2).mean()     # scalar mean of squares
>
>         beta = 1.0 - self.rate
>         self.ema_mean = beta * self.ema_mean + self.rate * batch_mean
>         self.ema_sqrs = beta * self.ema_sqrs + self.rate * batch_sqrs
>
>     def get_stats(self):
>         """
>         Return current (mean, std) after optional bias correction.
>         """
>         mean = self.ema_mean
>         variance = self.ema_sqrs - mean**2
>         std = max(self.limit, sqrt(max(0.0, variance)))
>
>         return mean, std
>
>     def __call__(self, batch, update=True):
>         if update:
>             self.update(batch)
>         return self.get_stats()
>
> def norm_res_img_obs(rec: jnp.array, true: jnp.array, norm: NormalizeMeanStd, training: bool):
>     # rec: reconstructed image, true: ground-truth image (both uint8)
>     res = f32(rec) - f32(true)
>
>     # independent Normalize object is used for each layer.
>     offset, scale = norm(res, training)
>
>     # approximately 95% of the values lie within 2 standard deviations
>     res = (res - offset) / scale / 4.0  # approximately 95% in [-0.5, 0.5]
>     # Apply dead zone to suppress near-zero noise
>     res = jnp.sign(res) * jnp.maximum(jnp.abs(res) - 0.05, 0)
>     # Map back to uint8
>     res = jnp.clip((res + 0.5) * 255, 0, 255).astype(np.uint8)
>     return res
> ```
>
> You may have noticed that the normalized residual image has deviated from the original distribution. The Normalized residual image added to the imaginary hint follows a dead-zone processed normal distribution in the pixel space. The motivation behind this design is as follows:
>
> 1. **Training stability**: Since the residual image needs to be learned and reconstructed in the upper-level world model, it is necessary to avoid its distribution drift during the training process as much as possible.
> 2. **Informative rather than photorealistic**: The residual corrected "image foresight" does not need to precisely match the future video distribution; instead, it mainly needs to be as informative as possible.
> 3. **Extreme compression in the encoder:** With the standard configuration (32 × 32-way categorical stochastic state), only a maximum of 128 bits of information can flow into the internal state each step. Therefore, even very blurry or stylized visual foresight is valuable as long as it carries task-relevant bits
>
> So far, we have thoroughly explained the details and motivation of residual image normalization.

---

> ### Author Response · Authors · 2025-11-22
> **Response to Reviewer asHi (Part 4)**
>
> ## Clarification on policy and rollout
>
> > W2.c: Clarification on the policy and temperature, the rollout horizon, whether hints are recomputed every step or cached, whether reconstructed rollouts computed from online or target decoder.
>
> Thank you for the very precise questions. We follow the official DreamerV3 hyperparameters and implementation details.
>
> |Question|Answer (fully aligned with official DreamerV3)|
> |-|-|
> |Policy temperature during training/evaluation|No temperature scaling is applied. Actions are sampled directly from the original policy distribution for both training and evaluation. The entropy regularization coefficient is fixed (as in Eq. 10 of DreamerV3).|
> |Temperature for open-loop imaginary rollouts|No temperature is used. The prior $\hat{z}_t^k$ is sampled from original stochastic state as Eq. 1.|
> |Rollout horizon H |H = 4 for every hierarchical layer every step of the encoding process.|
> |Are hints recomputed every real time step or cached?|Foresight hints (and residual predictions) are recomputed **every step** during encoding. No caching is performed.|
> |Online decoder or target decoder for reconstruction?|Foresight hints are decoded from online decoder. Target decoder is not involved. |
>
>
> Because the only new hyperparameter introduced by ResDreamer is the rollout horizon H, we have added a systematic sensitivity study (H = 4, 8, 16) on the continuous-visual-control benchmark DMC Vision [8] while keeping the total number of imagined frames fixed at 4. Results are provided in the global response "Additional Experiment Summary" and here: [https://anonymous.4open.science/r/ResDreamer-F33C/img/curves_score_dmc_vision.png](https://anonymous.4open.science/r/ResDreamer-F33C/img/curves_score_dmc_vision.png)
>
> ## Clarification on stop gradient operation
>
> > W2.d: Clarification on stop gradient operation.
>
> None of the computed and reconstructed observation  ($o_{\text{imag}}^k,  o_{\text{res}}^k,  \hat{o}_{\text{res}}^k$ ) pass gradients between layers. They merely transmit the pixel values of the error and image foresight.
>
> Thanks again for reminding us to clarify this matter. We have added stop gradient symbol in Figure 1, 2 and Eq. 4.
>
> ## Clarification on Compute budget
>
> > W2.e: Clarification on Compute budget.
>
> |**Compute budget**  |ResDreamer (50Mx2) |ResDreamer (100Mx2)|Dreamer V3 (100M)|
> |-|-|-|-|
> |**Total parameters**|92.0M|192.7M|109.5M|
> |**Foresight rollout step per environment step per layer**|4|4|0|
> |**Train ratio (Optimizer updates per environment step)**|32|32|32|
> |**Total wall-clock hours for 1M environment steps**|12.3|14.5|6.2|
>
>
> We aligned "Total parameters" and "Train ratio" to enable a fair comparison between ResDreamer and Dreamer V3 under the same computational budget. The additional total wall time of ResDreamer is derived from the foresight image obtained through visual grounded reasoning.
>
> ## Clarification on random seed and result aggregation
>
> > W2.f: Clarification on seeds and confidence interval.
>
> The experiments of all methods use random seeds 0,1,2,3, and the confidence interval is the standard deviation across all seeds. More specifically, for learning curves:
>
> - Each individual training run is divided into 40 uniformly spaced bins along the x-axis (environment steps).
> - The success rate within each bin is averaged to produce one data point per seed.
> - The final plotted curve shows the mean of these 40 points across the four seeds, and the shaded area represents standard deviation of the bin averages.
>
> Our experimental configuration with dreamer V3 is completely fair and aligned. The configurations of other baselines adopt the official default configuration. All code will be open source.
>
> ## Extended training for clearer success dynamics
>
> > W2.f: Why weren't Combat shulker, Combat wolf and Combat skeleton run up to more step for a more clear idea of the success dynamics.
>
> Thanks for the constructive feedback. We completely agree that these three tasks could be run longer for a clearer success dynamics.
>
> We have therefore retrained ResDreamer (50M×2) as well as a strong DreamerV3 ablation (single-layer DreamerV3 augmented with the same online-computed image foresight + reconstruction, but without hierarchical layers or residual modulation) for 2.5M environment steps.
>
> New curves:
>
> [https://anonymous.4open.science/r/ResDreamer-F33C/img/curves_success_2.4M.png](https://anonymous.4open.science/r/ResDreamer-F33C/img/curves_success_2.4M.png)
>
> Even with significantly longer training, ResDreamer continues to improve and substantially outperforms the enhanced DreamerV3 baseline. This further strengthens our original conclusions.
>
> We believe these additions and clarifications fully address your concerns. Thank you again for helping us make the experimental section much more rigorous and convincing!

---

> > ### Author Response · Authors · 2025-11-22
> > **Response to Reviewer asHi (Part 5)**
> >
> > ## Transformer based world models baseline
> >
> > > W3.1: Why were transformer based world models like IRIS not compared?
> >
> > We greatly appreciate this suggestion. To directly address it, we have adapted the official IRIS [1] codebase to MineDojo and run a faithful reproduction using three random seeds and the official default hyperparameters. The only necessary changes were:
> >
> > 1. Adding support for MineDojo's MultiDiscrete action space (instead of the original flat Discrete used on Atari).
> > 2. Setting MineDojo observations to 64×64 (instead of our usual 80×128) to match IRIS's hard-coded input resolution. This might weaken the model's capabilities, but is required for compatibility.
> >
> > Results (three seeds): [https://anonymous.4open.science/r/IRIS-Minedojo-2CB0/results/curves_iris.png](https://anonymous.4open.science/r/IRIS-Minedojo-2CB0/results/curves_iris.png)
> >
> > We have trained IRIS under default configuration for 7 days. Up to now, IRIS failed to achieve meaningful success even on the easiest Combat Spider task throughout 500K environment steps.
> >
> > IRIS [1] and other transformer based world models like TWM [2] is only officially tested on Atari environment. Our current conclusion regarding the reproduction of IRIS on Minedojo is that the default configuration is unable to complete the Minedojo task. Further hyperparameter tuning is still required.
> >
> > In contrast, the Dreamer V3 achieves or approaches the SOTA on over 150 tasks with a unified set of hyperparameters, including many 3D game environments. Therefore, we chose Dreamer V3 as the main baseline.
> >
> > We agree that a more competitive Transformer baseline would be valuable. The most promising direction, in our view, is to replace the RSSM encoder/decoder and sequence model in DreamerV3 with Transformer blocks while preserving DreamerV3's proven training recipe and hyperparameters. Unfortunately, this constitutes a non-trivial re-implementation that could not be completed within the rebuttal period. If the reviewers deem it necessary for the final version, we commit to adding such implementation (with full open-source code) in the public comments and the future revised paper.
> >
> > ## Comparing with Dino-WM and other pre-trained visual encoder based methods
> >
> > > W3.2: Why were pre-trained visual encoder based models like Dino-WM not compared?
> >
> > Thank you for this important question. After careful consideration, we conclude direct comparisons with Dino-WM is not feasible since Dino-WM has a different problem formulation. In Dino-WM:
> >
> > > (The Agent ) is asked to perform a sequence of actions $a_0, \cdots , a_T$ to reach the goal state $z_g$.
> >
> > > We train the models with our offline datasets without any reward. The planning cost is defined as the mean squared error (MSE) between the current latent state and the goal's latent state
> >
> > In contrast, **ResDreamer **is tested on online RL tasks. No offline datasets or expert trajectories is used. Furthermore, Dino-WM works with planning scenario with clearly defined success state or goal image. As **we focus on "dynamic tasks"**, the model needs to interact with dynamic objects, and the image of success/victory is trivial. It is not appropriate to define reward or cost in terms of the distance between a state and a goal representation.
> >
> > As for other pre-trained visual encoder-based methods, such as OpenVLA [9], the mainstream visual encoder DINOv2 ViT-L/14 used has 300M parameters just for the frozen visual encoder, which has exceeded the total parameter of our maximum model setup. This makes it difficult to align experimental configurations for fair comparisons.
> >
> > Further explanation of our baseline choices is included in Appendix D "Baseline Introduction".  Thank you again for the thoughtful comment!

---

> ### Author Response · Authors · 2025-11-22
> **Response to Reviewer asHi (Part 6)**
>
> ## Difference with other related hierarchical methods
>
> > W4: Clarification on difference with related hierarchical representations.
>
> Thank you for pointing out several highly relevant hierarchical works. We have studied them carefully and now provide a clear, structured comparison. The key distinctions are summarized below:
>
> |Hierarchical methods   |Application area  |Upward information channel     |Downward information channel  |Number of hierarchies|
> |-|-|-|-|-|
> |**ResDreamer** |Online RL  |Sensory reconstruction error |Visual foresight|Scalable|
> |**Hieros [3]**|Online RL|Latent representation sequence  |Subgoal reward |Scalable|
> |**Director [4]**  |Online RL|Latent representation |Subgoal representation and reward |2|
> |**Puppeteer [5]**|RL for Visual whole-body control for humanoids|None |Command sequence |2|
> |**STREAMER [6]**|Event segmentation|Latent representation |Latent representation vectors |Scalable|
> |**APC [7]**|Active visual perception and hierarchical planning|Latent representation prediction error   |Latent representation prediction|Scalable|
>
>
> How ResDreamer fundamentally differs:
>
> - Hierarchical WM has been used in the field of online RL [3, 4]. **Director [4]** only one lower level worker and one higher level manager. The manager gives feature space goals every K = 8 steps which decides reward of the worker. **Hieros [3]** makes multi-level scalable sub-goal planning possible. The abstract action generated by the upper layer is used as a discrete encoding sub-goal in the latent space of the lower layer, and is also trained under the RL paradigm. **Puppeteer [5]** directly uses expert trajectories from human MoCap data to train high-level goals for lower-level worker. Different from providing goal representation, the high-level **ResDreamer** layer modulates the visual-based planning representation with residual signal, helping the lower level encoder to obtain more informative visual information source.
> - In **APC [7]**, higher level predicts lower latent vector and visible prediction error for latent feature. APC can perform planning and model predictive control without reconstructing sensory signal. Hierarchical planning based on APC shows adaptability to goal changes during training, while its RL baseline performance is seriously dropped. However, the hierarchical planning and decision-making capabilities of APC have only been verified in the 2D maze navigation environment.
> - **STREAMER [6]** is a self-supervised approach for hierarchical representation learning and is tested on video events segregation. In STREAMER, each layer can access the latent representation of adjacent layers and predict the future. The time window is dynamically adjusted based on whether the prediction error is higher than the threshold. ResDreamer is different from STREAMER in information flow design and time window management methods. We believe that supporting dynamic time window management in future work will further improve the practicability of ResDreamer.
>
> Thank you again for your valuable time. I feel honored that you could examine our paper in such detail.
> If you have any follow-up questions or suggestions, we will be happy to respond further.
>
> ## *References*
>
> [1] Micheli, V., Alonso, E., & Fleuret, F. (2022). Transformers are sample-efficient world models. *arXiv preprint arXiv:2209.00588*.
>
> [2] Robine, J., Höftmann, M., Uelwer, T., & Harmeling, S. Transformer-based world models are happy with 100k interactions, 2023. *URL *[*https://arxiv*](https://arxiv)*. org/abs/2303.07109*.
>
> [3] Mattes, P., Schlosser, R., & Herbrich, R. (2023). Hieros: Hierarchical imagination on structured state space sequence world models. *arXiv preprint arXiv:2310.05167*.
>
> [4] Hafner, D., Lee, K. H., Fischer, I., & Abbeel, P. (2022). Deep hierarchical planning from pixels. *Advances in Neural Information Processing Systems*, *35*, 26091-26104.
>
> [5] Hansen, N., SV, J., Sobal, V., LeCun, Y., Wang, X., & Su, H. (2024). Hierarchical world models as visual whole-body humanoid controllers. *arXiv preprint arXiv:2405.18418*.
>
> [6] Mounir, R., Vijayaraghavan, S., & Sarkar, S. (2023). Streamer: Streaming representation learning and event segmentation in a hierarchical manner. *Advances in Neural Information Processing Systems*, *36*, 45694-45715.
>
> [7] Rao, R. P., Gklezakos, D. C., & Sathish, V. (2023). Active predictive coding: A unifying neural model for active perception, compositional learning, and hierarchical planning. *Neural Computation*, *36*(1), 1-32.
>
> [8] Ortiz, J., Dedieu, A., Lehrach, W., Guntupalli, J. S., Wendelken, C., Humayun, A., ... & Murphy, K. P. (2024). DMC-VB: A Benchmark for Representation Learning for Control with Visual Distractors. *Advances in Neural Information Processing Systems*, *37*, 6574-6602.
>
> [9] Kim, M. J., Pertsch, K., Karamcheti, S., **ao, T., Balakrishna, A., Nair, S., ... & Finn, C. (2024). Openvla: An open-source vision-language-action model. *arxiv preprint arxiv:2406.09246*.

---

### Official Review · Reviewer_KNWC · 2025-11-04

**Soundness:** 3
**Presentation:** 3
**Contribution:** 3
**Rating:** 6
**Confidence:** 2

**Summary:**

The paper introduces ResDreamer, a hierarchical world model that connects multiple Dreamer-style world models via residual signals and imaginary hints. Each layer processes enhanced visual observations that include residual errors from the lower level and imagined future frames. The approach is motivated by predictive coding and aims to build scalable, biologically inspired world models that improve efficiency in complex, open-ended RL tasks such as combat scenarios in MineDojo.

**Strengths:**

* The proposed hierarchical residual connection mechanism is novel and grounded in predictive coding theory.
* The experimental results suggest improved sample efficiency and task success rates compared to DreamerV3 and other baselines.
* The document has a well-researched and -written literature review in model-based RL and hierarchical representation learning.

**Weaknesses:**

1. While the concept of “imaginary hints” — visual foresight reconstructed from imagined trajectories — is intriguing, it is not well explained how this component quantitatively drives the observed performance gains.
2. How do the hints differ from the standard Dreamer imagination rollout?
3. Why does adding residual-corrected foresight lead to such large improvements?
4. What is the exact ablation result isolating the effect of imaginary hints versus residual feedback?
5. The text claims that imaginary hints correspond to “dynamic CNN kernels” and “gaze control,” but the analysis in Figure 5 (“Only residual hints”) is the only direct test of this component, yet the explanation of why it performs better is purely qualitative. A more systematic analysis (e.g., measuring predictive accuracy of imagined trajectories or visualizing hint quality over time) would help substantiate the claim.
6. Since the model introduces several interacting ideas (hierarchical residuals, imaginary hints, layer conditioning), it’s difficult to disentangle their contributions from each other.

**Questions:**

1. Could you clarify how the imaginary hint differs from Dreamer’s latent imagination in implementation?
2. Is the imaginary hint computed online or only during training?
3. Did you experiment with removing the residual correction to see if the improvement persists?

---

> ### Author Response · Authors · 2025-11-22
> **Response to Reviewer KNWC (Part 1)**
>
> Dear Reviewer KNWC,
>
> We sincerely appreciate your thorough and insightful review of our manuscript. To address your concerns regarding disentangling the contributions of each interacting component, we have conducted additional supplementary experiments. We hope that our revised manuscript and the following responses adequately clarify your questions and resolve most of your concerns.
>
> ## Clarification on how the imaginary hint differs from Dreamer
>
> > Q1 (W2):  Clarification on how the imaginary hint differs from Dreamer's latent imagination.
>
> Dreamer uses latent imagination only for **actor-critic** training. The **encoder** and **decoder** of Dreamer's world model (WM) only input and reconstruct the original environmental observations. Our **actor-critic** training method is the same as that of Dreamer, but the **encoder** and **decoder** additionally process the low-level visual reconstruction residuals.
>
> In Dreamer, the **decoder** only drives the representation learning of WM through the reconstruction loss. We enable the agent to **envision the future** with residual enhanced visual hint by additionally leveraging WM's ability to reason and reconstruct.
>
> |**Aspect**|Dreamer |**Ours**|
> |-|-|-|
> |**Actor training**|Latent imagination trajectory|Same as Dreamer|
> |**Critic training**|Latent imagination trajectory & environment interaction trajectory|Same as Dreamer|
> |**Encoder input (computed at WM training & policy inference )**|Environment observation|Environment observation &  lower level residuals & imaginary hint|
> |**Decoder output (computed at WM training & policy inference)**|Environment observation reconstruction |Environment observation & lower residual reconstruction |
> |**Consequence**|Representation learning is driven solely by real frame reconstruction and reward modeling |Representation learning is additionally supervised by lower-layer residual and informed by residual-corrected image foresight|
>
>
> In short: Dreamer's encoder never "sees" anything from its own imagination—only real past and present frames.  ResDreamer lets encoder directly access more informative reasoning representation.
>
> ## Clarification on the online computation of imaginary hint
>
> > Q2:  Clarification on when the imaginary hint is computed.
>
> The imaginary hint, as a observation enhancement, is computed whenever the encoder is used. From another perspective, imaginary hints can be viewed as using ResDreamer's reasoning capabilities for observation preprocessing or enhancement.
>
> Since foresight hints are computed every step of the encoding process, it is used both in training and policy inference. The pseudocode for strategy reasoning and world model training is elaborated in Algorithm 1, 2 in Appendix A.
>
> ## Add ablation on residual connection
>
> > Q3(W4, W5):  Add ablation with residual correction removed.
>
> Thank you for the constructive suggestion. We strongly agree that ablation experiments on residual correction are necessary. We have directly followed your advice and added the two critical ablations:
>
> 1. ResDreamer (Rollout hints only, residual connections removed): ResDreamer still performs image rollout, but only use the unmodulated imaginary hint. Actor-critical and other prediction heads use the learned representation simply by accessing the latent vector class of all layers. Ablation curve: [https://anonymous.4open.science/r/ResDreamer-F33C/img/curves_success_ablation.png](https://anonymous.4open.science/r/ResDreamer-F33C/img/curves_success_ablation.png)
> 2. Dreamer (With image foresight, no hierarchy, no residual): A strong single-layer DreamerV3 baseline augmented with the same image foresight mechanism, but without hierarchical layers or residual modulation. It's included in ablation curve as well as a 2.5M environment-steps additional setup:  [https://anonymous.4open.science/r/ResDreamer-F33C/img/curves_success_2.4M.png](https://anonymous.4open.science/r/ResDreamer-F33C/img/curves_success_2.4M.png)
>
> All experimental configurations are aligned with the model analysis in the the paper. Please refer to the latter part of the rebuttal: **Disentangle contributions from hierarchy, lookahead rollout and residual connection** for more detailed explaination and analysis.

---

> > ### Author Response · Authors · 2025-11-22
> > **Response to Reviewer KNWC (Part 2)**
> >
> > ## Explanation of significant improvements
> >
> > > W3: Why does adding residual-corrected foresight lead to such large improvements?
> >
> >  In one sentence:
> > **Residual-modulated image foresight is substantially more informative than either the raw reconstructed image or the residual alone, because upper layers explicitly model previously unmodeled residuals and thus provide task-relevant future visual signals.**
> >
> > On the one hand, the feedforward channel provides modulated foresight image which makes the observation source move informative. On the other hand, the feedback channel allows error propagation upward, resulting in a more comprehensive inner world representation.
> >
> > Specifically, in "Combat shulker": ResDreamer (100Mx2) is the only method that solves combating a shulker in $1\times 10^6$ environment steps. The Shulker launches a guided projectile that causes prolonged levitation and causes fall damage afterwards. This poses complex and challenging dynamic interaction mechanisms for the agent. We found that the residual-modulated visual reasoning representation provided significant assistance in understanding this dynamic interaction mechanism.
> >
> > ## Explanation of how visual foresight quantitatively drives the performance gains
> >
> > > W1: While the concept of "imaginary hints" is intriguing, it is not well explained how this component quantitatively drives the observed performance gains.
> >
> > Thank you for this excellent and crucial comment. We fully agree that the paper originally lacked rigorous quantitative evidence linking the design of visual foresight to the observed gains. We have now added a sensitivity study that directly measures how horizon length quantitatively affect performance.
> >
> > In our added experiment: **Foresight Horizon Sensitivity Analysis**, we further quantitatively verified how theforesight horizon quantitatively affects the improvement of information volume. We compare ResDreamer with different foresight horizon ($H=4, 8, 16$) on DeepMind Control Vision continuous control suite. The total number of frames for visual foresight has been aligned to 4 frames ($H=8$ has stride $D=2$, and $H=16$ has stride $D=4$). Please refer to the Global Response: **Additional Experiments Summary** for more details. Score curves: [https://anonymous.4open.science/r/ResDreamer-F33C/img/curves_score_dmc_vision.png](https://anonymous.4open.science/r/ResDreamer-F33C/img/curves_score_dmc_vision.png)
> >
> > Generally, foresight with longer horizon and larger stride is more informative, but the convergence speed may be slower in some cases.
> >
> > In "DMC Reacher Hard" which requests  precise control of a two-link robotic arm, all ResDreamer configurations converge to the full score within 1M steps. The reasoning table with a shorter time window converges the fastest.
> >
> > In "DMC Hopper Hop" which requires a 4-joint legged robot to perform a high-speed forward leap while maintaining balance, agent with rollout horizon $H=16$ significantly outperforms other setup for 16 steps roughly covers the entire cycle of a single jump.
> >
> > These results clearly demonstrate that performance is directly controlled by how well the foresight horizon matches the temporal extent of the task-relevant dynamics. Longer strides sacrifice short-term density but capture farther future events that are critical for tasks with extended temporal dependencies (e.g., anticipating landing after a hop).

---

> ### Author Response · Authors · 2025-11-22
> **Response to Reviewer KNWC (Part 3)**
>
> ## Disentangle contributions from hierarchy, lookahead rollout and residual connection
>
> > W6: Since the model introduces several interacting ideas (hierarchical residuals, imaginary hints, layer conditioning), it's difficult to disentangle their contributions from each other.
>
> Thank you for this critical comment. We think disentangle the contributions from each design makes our work much more solid. We have now completed the full combinatorial study you requested (all runs with identical hyper-parameters). The results are summarized in the table below
>
> #### **Table: Ablation result**
>
> |#|Configuration|Hierarchy|Rollout Hint|Residual Connection|Actor/Critic Input|Success Rate|
> |-|-|-|-|-|-|-|
> |1|ResDreamer (50M×3)|3 layers|✓|✓|Bottom layer only|0.776|
> |2|ResDreamer (50M×2) – main model|2 layers|✓|✓|Bottom layer only|0.727|
> |3|ResDreamer (Residual-only, no hint)|2 layers||✓|Bottom layer only|0.563|
> |4|DreamerV3 (50M) + image foresight (no hierarchy)|1 layer|✓||Single layer|0.559|
> |5|ResDreamer (Hint-only, no residual)|2 layers|✓||Concatenated latents from all layers|0.400|
> |6|ResDreamer (Heads conditioned on all)|2 layers|✓|✓|Concatenated latents from all layers|0.377|
>
>
> - Config 1 & 2 proves the scalibility of ResDreamer.
> - Config 3 & 4 show removing either the hint or the residual causes massive drops. The residual alone is insufficient, and the hint alone is not as helpful.
> - Config 5 & 6 demonstrate conditioning actor-critic on concatenated latents from all layers leads to poor performance. This confirms that before the world model fully converges, unstable high-level latent representations can slow down RL convergence. In contrast, residual-modulated image foresight provides relatively stable reasoning representations.
>
> The core experimentally verified insight is: **residual-modulated image foresight provides strictly more useful information to the encoder than any of its partial components.**
>
> ---
>
> Thank you again for your constructive and detailed feedback, which helped us greatly improve our work. We hope these response and additional experiments adequately address your concerns. We are happy to provide additional clarifications, experiments, or discussions upon request.

---

### Author Response · Authors · 2025-11-22
**Additional Experiment Summary**

We sincerely thank all reviewers for their constructive feedback.

- Reviewers **KNWC** & **q9v4** suggested adding of residual signal ablation and using only the unmodulated image foresight.
- Reviewer **q9v4** suggested adding sensitivity analysis on foresight horizon.
- Reviewer **asHi** mentioned longer environment steps setup.
- Reviewers **eFj2** & **q9v4** is interest in evaluation on wider range of tasks.
- Reviewers **eFj2** & **asHi** mentioned other hierarchical baseline such as IRIS [1].

To thoroughly address the reviewers' concerns, we have conducted the following new experiments.

## Additional Experiment Setup

1. ResDreamer (50Mx2, rollout hints only, no residual connections): We remove all residual connections from the standard ResDreamer architecture. The actor, critic, and all auxiliary prediction heads directly access the latent representations from all layers. This ablation isolates the contribution of residual corrections.
Anonymous link for results: [https://anonymous.4open.science/r/ResDreamer-F33C/img/curves_success_ablation.png](https://anonymous.4open.science/r/ResDreamer-F33C/img/curves_success_ablation.png)
2. ResDreamer (50Mx2, extended to 2.5M environment steps): To reveal clearer learning dynamics and rule out insufficient training as a factor, we extend training of the three combat tasks (Shulker, Wolf, Skeleton) to 2.5M environment steps.
Success curves: [https://anonymous.4open.science/r/ResDreamer-F33C/img/curves_success_2.4M.png](https://anonymous.4open.science/r/ResDreamer-F33C/img/curves_success_2.4M.png)
3. ResDreamer (12Mx2, Rollout horizon H=4, 8, 16 on DMC Vision ): We evaluate different rollout horizons on the continuous visual control benchmark DMC Vision [2]. To maintain a fixed visual foresight budget, we use only 4 imagined frames in total. This leads to stride D=1 (H=4), D=2 (H=8), and D=4 (H=16). We select the 5 most difficult tasks that Dreamer V3 did not converge to full score.
Score curves: [https://anonymous.4open.science/r/ResDreamer-F33C/img/curves_score_dmc_vision.png](https://anonymous.4open.science/r/ResDreamer-F33C/img/curves_score_dmc_vision.png)
4. DreamerV3 (50M parameters) + image foresight (no residual modulation):  We augment the original single-layer DreamerV3 with the same visual grounding and image foresight mechanism used in ResDreamer, but without hierarchical layers or residual corrections.
Ablation curves: [https://anonymous.4open.science/r/ResDreamer-F33C/img/curves_success_ablation.png](https://anonymous.4open.science/r/ResDreamer-F33C/img/curves_success_ablation.png)
Extended 2.5M-step curves:  [https://anonymous.4open.science/r/ResDreamer-F33C/img/curves_success_2.4M.png](https://anonymous.4open.science/r/ResDreamer-F33C/img/curves_success_2.4M.png)
5. IRIS baseline: We reproduce the official implementation of "Transformers are Sample-Efficient World Models" [1] with its default hyperparameters. Up to now (500K environment steps / ~7 days training).
Success curves: [https://anonymous.4open.science/r/IRIS-Minedojo-2CB0/results/curves_iris.png](https://anonymous.4open.science/r/IRIS-Minedojo-2CB0/results/curves_iris.png)

## Conclusion from original and new experiments

- ResDreamer achieves superior sample efficiency with fewer parameters compared to baselines.
- The combination of hierarchical structure, hint rollout, and residual modulation is significantly more effective than any subset of these components.

*References*

[1] Micheli, V., Alonso, E., & Fleuret, F. (2022). Transformers are sample-efficient world models. *arXiv preprint arXiv:2209.00588*.

[2] Ortiz, J., Dedieu, A., Lehrach, W., Guntupalli, J. S., Wendelken, C., Humayun, A., ... & Murphy, K. P. (2024). DMC-VB: A Benchmark for Representation Learning for Control with Visual Distractors. *Advances in Neural Information Processing Systems*, *37*, 6574-6602.

---

### Meta-Review · Area_Chair_fp3M · 2025-12-26

**Summary:**

The paper proposes ResDreamer - a hierarchical world model for RL that adds to Dreamer a layered approach of processing reconstruction errors from lower layers.

Concerns were raised about some ablations and baselines (which the authors addressed in their rebuttal), and about the presentation. The presentation, even after the authors attempted clarification, is very difficult to understand and significantly below the bar of a top-tier conference. I suggest the authors to work on the paper’s presentation and try to elucidate the main idea and explain it better, focusing on how it connects to the technical implementation.

**Reviewer Concerns:**

detailed above.

**Reviewer Scores:**

6,4,4,4

The decision is based on the reviews and on my own review of the paper.

---

### Decision · Program_Chairs · 2026-01-26

Reject